# Oxidative stress causes a reversible decrease of deubiquitylases activity in old vertebrate brains

Amit Kumar Sahu [1,6], Alberto Minetti[1,7], Domenico Di Fraia[1,6], Antonio Marino[1,8], Patrick Rainer Winterhalter[2], Daniela Giustarini[3], Ranieri Rossi[3], Andreas Simm[2], Francesco Neri [1,4], Federico Galvagni [3], Christoph Gerhardt [5], Thorsten Pfirrmann [5] ✉ & Alessandro Ori [1,9] ✉

The ubiquitin–proteasome system is essential for neuronal proteostasis, yet its function declines with age. How aging affects deubiquitylating enzymes (DUBs) in the vertebrate brain remains unclear. Here we used activity-based proteomics to profile cysteine protease DUBs in aging mouse and killifish brains. We identified a subset of DUBs that progressively lose catalytic activity with age despite stable protein abundance. Mechanistically, oxidative stress impaired DUB function through thiol oxidation, whereas antioxidant treatment with N-acetylcysteine ethyl ester (NACET) restored activity in aging brains. In human iPSC-derived neurons, global DUB inhibition and targeted inhibition of USP7, one of the most strongly age-affected DUBs, partially recapitulated ubiquitylation changes observed in aged brains. Temporal analysis in mice further revealed that DUB inhibition precedes proteasome decline during brain aging. Together, these findings identify redox-sensitive DUBs that lose activity with age and suggest impaired deubiquitylation as an early, potentially reversible driver of proteostasis decline in the aging brain.

Aging is characterized by widespread functional and molecular changes in the brain, including synaptic dysfunction, increased oxidative stress, and altered protein homeostasis (proteostasis)[1]. Among the cellular mechanisms governing proteostasis, the ubiquitin-proteasome system (UPS) plays a central role in signaling, stress responses, and protein degradation by attaching ubiquitin to lysine residues of specific target proteins[2]. Alterations in proteasome activity and ubiquitylome have been well characterized in the context of aging in both invertebrates and vertebrates[2–6]. Within the UPS, ubiquitin ligases and deubiquitylases (DUBs) act antagonistically to modulate protein fate and signaling pathways dynamically.

Among these enzymes, DUBs constitute a diverse family of more than 100 enzymes identified in vertebrates, broadly classified into cysteine proteases and zinc-dependent metalloproteases[7]. DUBs are central regulators of the cell cycle, immune signaling, and control multiple aspects of neuronal function, including mitochondrial quality control, synaptic plasticity, and stress responses[7]. Altering DUB activity has been linked to lifespan in nematodes[4,8], and dysregulation of

[1]Leibniz Institute on Aging - Fritz Lipmann Institute (FLI), Jena, Germany. [2]Clinic for Heart Surgery (UMH), Martin-Luther-University Halle-Wittenberg, Halle (Saale), Germany. [3]Department of Biotechnology, Chemistry and Pharmacy, University of Siena, Siena, Italy. [4]Department of Life Sciences and Systems Biology, University of Turin, Torino, Italy. [5]Institute for Molecular Medicine, Department of Medicine, Health and Medical University Potsdam, Potsdam, Germany. [6]Present address: Cologne Excellence Cluster for Cellular Stress Response in Aging-Associated Diseases (CECAD), University of Cologne, Cologne, Germany. [7]Present address: Department of Neurosurgery, University Hospital Erlangen, Friedrich-Alexander University Erlangen Nuremberg, Erlangen, Germany. [8]Present address: Proteomics Research Infrastructure, University of Copenhagen, Copenhagen, Denmark. [9]Present address: Genentech Inc., South San Francisco, CA, USA. ✉e-mail: thorsten.pfirrmann@hmu-potsdam.de; alessandro.ori@leibniz-fli.de

specific DUBs in humans leads to several neurodegenerative diseases, such as spinocerebellar ataxia and Parkinson's disease[9-12]. Emerging evidence has implicated the tight regulation of DUB activity by protein binding and post-translational modifications, including the redox regulation of the cysteine protease DUBs[7,13]. However, a systematic understanding of how DUB functions is altered in the aging brain, the mechanisms driving these changes, and the consequences of altered DUB activity at the molecular level are still lacking.

To fill this gap, we profiled the activity of cysteine-dependent DUBs in mouse (*Mus musculus*) and killifish (*Nothobranchius furzeri*) brains across different age groups. Using activity-based probes coupled with mass spectrometry and biochemical validation, we found that the average DUB activity is reduced by approximately 40% in old brains due to increased cysteine oxidation, and it can be restored by antioxidant treatments both in vitro and in vivo. By employing human-induced pluripotent stem cell (iPSC)-derived neurons (iNeurons), we demonstrate that DUB inhibition can recapitulate a significant fraction of age-related ubiquitylated signatures identified in the mouse brains. Importantly, we show that decreased DUB activity temporally precedes impairment of proteasome activity and accumulation of ubiquitylated proteins during brain aging. Together, our findings revealed a conserved susceptibility of DUBs to redox imbalance and provided insights into impaired deubiquitylation as a contributing factor to age-associated proteostasis decline.

## Results

### Global decrease in active deubiquitylases in aging brains

To understand how the activity of DUBs changes during the aging process in mouse brains, we performed a DUBs activity assay by utilizing a fluorescent deubiquitylase substrate, as previously used in ref. 14. We observed a decrease in the global DUB activity in old (30 months) mouse brain lysates compared to young (3 months) ones (Fig. 1A). To gain deeper insight into the activity of individual DUBs, we used activity-based probes to enrich cysteine protease DUBs and their identification using a mass spectrometry (MS) approach. To maximize the capture of individual DUBs, we pooled probes from three different classes that included a mono-ubiquitin recognition element specific for DUBs, a biotin-tag for DUBs' pulldown, and either a propargylamide (PA), a vinylmethylester (VME), or a vinylsulfone (VS) electrophilic warhead to covalently interact with the nucleophilic cysteine residue in the active sites of DUBs[15-17] (Fig. 1B). Hereafter, we referred to the pool of activity-based probes as DUB probes. As a negative control, we pre-treated brain lysates with N-ethylmaleimide (NEM) that can irreversibly alkylate any free sulfhydryl group, such that DUB probes will fail to bind and enrich DUBs in the pulldown assay (Fig. 1B, Fig. S1A). We analyzed a first cohort of samples comprising brains from 3 young (3 months old) and 3 old (33 months old) C57BL/6J male mice. Principal Component Analysis (PCA) of enriched proteins depicted a clear separation based on age groups and DMSO (vehicle) or NEM treatment, despite some variability among old NEM samples (Fig. 1C).

Using this approach, we identified a total of 56 distinct DUBs across all age and treatment conditions (Supplementary Data 1). We defined the quantity of DUBs being enzymatically active as the abundance of DUBs enriched over the NEM control. This revealed the significant enrichment (absolute average $\log_2$ FC > 0.58; Qvalue < 0.05) of 44 and 45 active DUBs in young and old brains, respectively (Fig. S1B). Even with stringent washing steps, several known DUB substrates and interactors were co-enriched, likely due to their physical association with DUBs, as previously reported[18,19] (Fig. 1D, Fig. S1C). Consistent with the fluorescent DUB substrate-based assay, we observed an overall decrease in the activity of 18 out of 56 detected DUBs in old brains compared to young ones (Fig. 1E, F). This decline was independently validated in a second experimental cohort, which revealed reduced activity in 20 out of 54 detected DUBs, thereby confirming the robustness of our findings (Fig. S1D, Fig. 1F). Among significantly

affected DUBs, we identified UCHL1 (a neuron-specific DUB) and YOD1 (a DUB broadly expressed across brain cell types) that have been linked to neurodegenerative diseases[20-25] (Fig. 1F). On the other hand, the activity of some other neurodegeneration-associated DUBs, such as CYLD and UCHL3[26-28], was not altered by aging (Fig. 1F).

We have previously shown that changes in protein ubiquitylation and decreased proteasome activity are conserved aging signatures in killifish and mice[3,5,29]. To understand whether the decrease in DUB activity during brain aging is also a conserved phenotype across vertebrate species, we repeated our DUB probe-based activity profiling with brain lysates of killifish of different age groups. Consistent with our mouse data, we observed a decrease in global DUB activity in old killifish brains compared to the young ones (Fig. S1E, Supplementary Data 1).

Importantly, in both species, the age-dependent decline in DUB activity was largely independent of corresponding changes in DUB protein abundance (Fig. 1F, Fig. S1F). In mouse brains, 20 out of 27 DUBs that exhibited age-dependent changes in activity in at least one cohort showed no significant alteration in protein abundance (absolute $\log_2$ FC > 0.58 and Pvalue < 0.05) (Fig. 1F). Similarly, in aging killifish brains, 5 out of 6 DUBs displaying reduced activity did not show changes in protein abundance, with the exception of USP25, which exhibited decreased protein levels (Fig. S1F). For a subset of DUBs whose total protein abundance was not detected (7 in mouse and 1 in killifish), it was not possible to assess whether the observed activity changes were independent of protein abundance.

Together, these findings demonstrate that DUB activity declines with age in both mouse and killifish brains and that this reduction is often uncoupled from DUB abundance changes, suggesting that post-translational mechanisms contribute to regulating DUB activity during brain aging.

### Thiol oxidation reduces deubiquitylase activity in aging brains

Elevated levels of reactive oxygen species (ROS), along with chronically activated antioxidant defense mechanisms that may become insufficient over time, have been implicated in aging and age-associated diseases, contributing to impaired protein homeostasis and reduced enzymatic function[1,30]. Consistent with this, analysis of proteome data from aged mouse brains[5] revealed increased abundance of several antioxidant-associated proteins, including apolipoprotein D (APOD) and selenoprotein K (SELENOK), alongside reduced abundance of protective factors such as serine protease inhibitor A3K (SERPINA3K), reactive oxygen species modulator 1 (ROMO1), and haptoglobin (HP) (Fig. S2A). Notably, we also observed a decline in nuclear factor erythroid 2-related factor 2 (NRF2), the master regulator of antioxidant gene expression and glutathione (GSH) biosynthesis, accompanied by reduced GSH levels in aging mouse brains (Fig. S2B, C).

Since many DUBs rely on the thiol of cysteine residues at their active site to cleave ubiquitin molecules from substrates or another ubiquitin molecule, we hypothesized that oxidative stress-mediated thiol oxidation in the brains of old mice might explain the reduced enrichment of active DUBs in our experiments[31]. To test this hypothesis, we quantified the concentration of reduced form of thiols (-SH) in young and old mouse brains' lysate using Ellman's reagent, 5,5'-dithio-bis-(2-nitrobenzoic acid) (DTNB), which is also used to study cysteine oxidation[32,33]. We observed a reduction in thiol groups in old brain lysates compared to young ones in two independent experiments, suggesting that thiol oxidation increases with age (Fig. 2A). Further, to understand whether the DUB activity depends on the concentration of reduced thiol groups in cysteines, we performed a linear regression analysis. We observed a statistically significant linear relationship (Pvalue = 0.0043, $R^2$ = 0.45), confirming that thiol concentration (or, in other words, cysteine's oxidation) correlates significantly with DUB activity in the same set of samples (Fig. S2D).

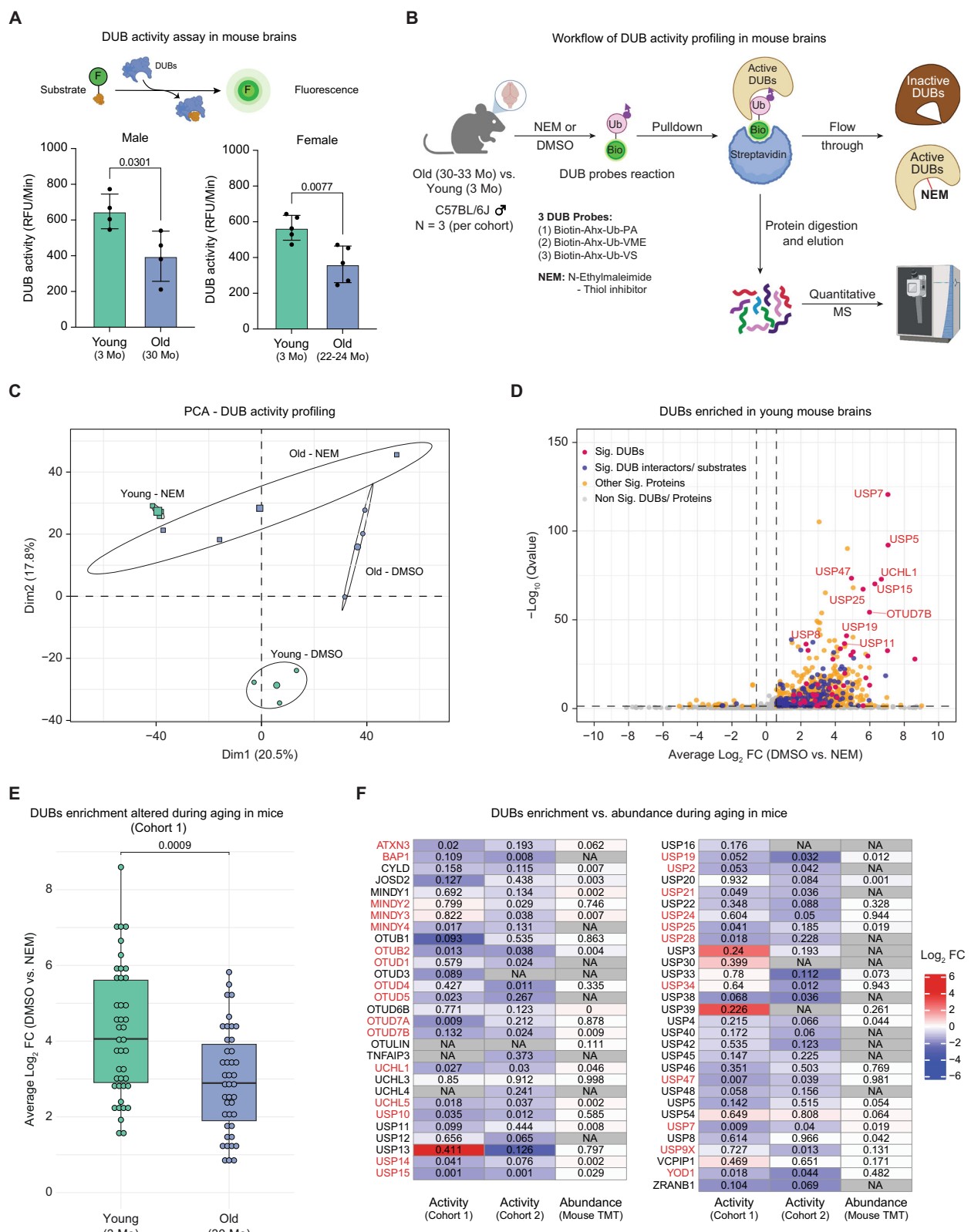

**A** DUB activity assay in mouse brains

**B** Workflow of DUB activity profiling in mouse brains

**C** PCA - DUB activity profiling

**D** DUBs enriched in young mouse brains

**E** DUBs enrichment altered during aging in mice (Cohort 1)

**F** DUBs enrichment vs. abundance during aging in mice

Next, we asked whether we could reverse the effect of oxidation on the activity of DUBs in old brain lysates. We treated brain lysates from old mice with dithiothreitol (DTT), a reducing reagent previously shown to enhance DUB activity in vitro[13], followed by DMSO or NEM and DUB probe-based enrichment, as in (Fig. 1B). We observed a significant increase in the overall enrichment of active DUBs in old brain lysates treated with DTT compared to the non-treated lysate (Fig. 2B,

Fig. S2E, Supplementary Data 1). Further, by performing the relative enrichment analysis of the protein quantity, we found an enhancement of individual DUB activity, such as that of UCHL1 and UCHL5, in DTT-treated old brain lysates. However, DTT treatment didn't significantly alter the activity of other DUBs, e.g., YOD1 and OTUD5 (Fig. 2C). On the other hand, DTT did not impart significant changes in the enrichment of active DUBs in young brain lysates (Fig. S2F, G). These observations

**Fig. 1 | DUB activity decreases during mouse brain aging. A** Top: schematic of fluorescent substrate–based measurement of DUB activity. Bottom: DUB activity in young (3 months) and old (30 months) C57BL/6J male mice (left, $N = 4$) and young (3 months) and old (22–24 months) C57BL/6J female mice (right, $N = 5$) brain lysates. Two-tailed unpaired t-test with Welch's correction. RFU = relative fluorescence units. Data are shown as mean ± SD. **B** Workflow for enrichment and identification of active DUBs using ubiquitin-based activity probes in young (3 months) and old (30–33 months) C57BL/6J male mouse brains ($N = 3$). Ub, mono-ubiquitin; Bio, biotin tag; arrowheads indicate probe warheads (propargylamide (PA), vinyl methyl ester (VME), or vinyl sulfone (VS)). **C** PCA of proteins enriched by DUB probes from mouse brains (cohort 1). Ellipses represent 95% confidence intervals for DMSO or NEM treated young and old samples. Percent variance explained by each principal component is indicated ($N = 3$). **D** Volcano plot of proteins enriched from young mouse brains (cohort 1, $N = 3$). Vivid pink dots indicate active DUBs; deep indigo dots denote known DUB substrates and interactors[18,19]; vivid yellow dots indicate other significantly co-enriched proteins; light gray dots represent non-significant proteins. Seven non-DUB proteins were excluded for clarity.

Differential abundance was assessed using Spectronaut (Qvalue). Dashed lines indicate thresholds (absolute average $\log_2$ fold change (|average $\log_2$ FC |) > 0.58; Qvalue < 0.05). **E** Boxplot showing DUB enrichment in young (3 months) and old (30 months) mouse brains (cohort 1). 42 DUBs detected in both groups are shown ($N = 3$; |average $\log_2$ FC | > 0.58; Qvalue < 0.05; two-tailed Wilcoxon rank-sum test). Data are shown as the median (central line) and the interquartile range (25–75th percentiles, box limits). Whiskers extend to 1.5× the interquartile range, and individual measurements of DUB enrichment are overlaid as dots. **F** Heatmap comparing age-associated changes in DUB activity (this study; cohort 1: 30 vs. 3 months; cohort 2: 33 vs. 3 months; $N = 3$) and protein abundance (TMT proteomics from[5]; 33 vs. 3 months). Values represent Pvalues from two-tailed unpaired t-tests with Welch's correction. 'NA' indicates DUBs not detected. DUBs highlighted in red exhibit significant (Pvalue < 0.05) age-associated activity changes in at least one cohort. N refers to the number of biological replicates used in the experiment. Related to Supplementary Data 1. Source data are provided as a Source Data file. Created in BioRender. Sahu, A. (2026) https://BioRender.com/5q3eb5x.

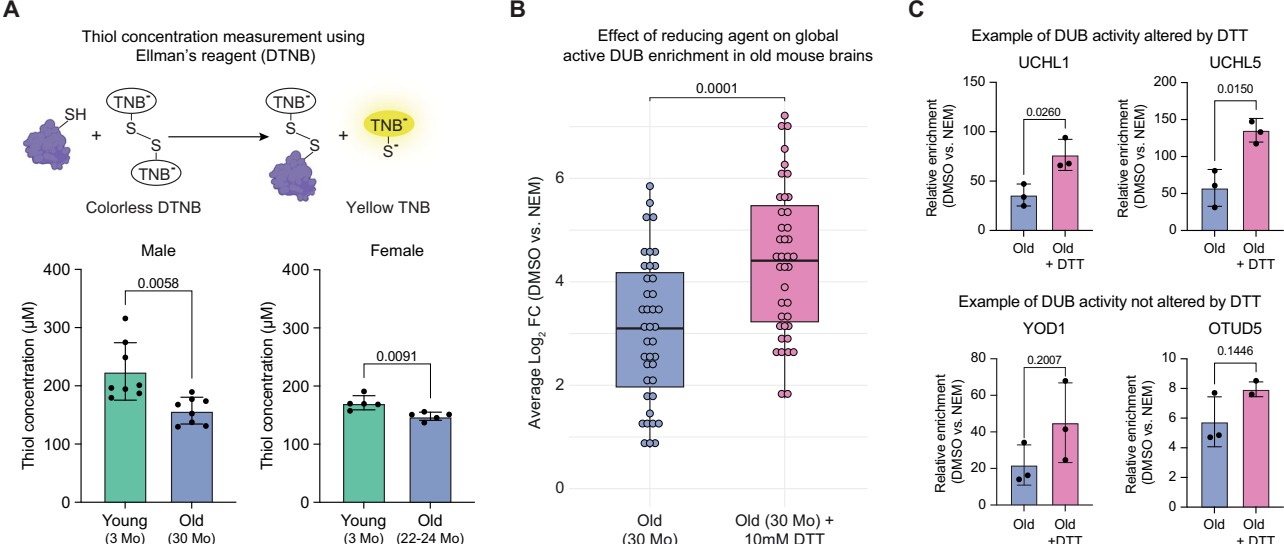

**Fig. 2 | Increased thiol oxidation causes a reversible decline in DUB activity during mouse brain aging. A** Top: Schematic illustrating Ellman's reagent (DTNB)–based detection of reduced thiol groups, producing the yellow TNB product measurable at 412 nm via spectrophotometry. Bottom: Reduced thiol concentrations in brain lysates from young (3 months) and old (30 months) C57BL/6J male mice (left, $N = 8$) and young (3 months) and old (22–24 months) C57BL/6J female mice (right, $N = 5$). Two-tailed unpaired t-test with Welch's correction. RFU = relative fluorescence units. Data are shown as mean ± SD. **B** Boxplot of individual active DUB enrichment in old mouse brain lysates with or without 10 mM DTT treatment. 40 DUBs detected under both conditions are shown (N = 3; |average

$\log_2$ FC | > 0.58; $Q$ < 0.05; two-tailed Wilcoxon rank-sum test). Data are shown as the median (central line) and the interquartile range (25–75th percentiles, box limits). Whiskers extend to 1.5× the interquartile range, and individual measurements of DUB enrichment are overlaid as dots. **C** Bar plots showing representative cysteine protease DUB enrichment in old mouse brain lysates treated with or without 10 mM DTT ($N = 3$; two-tailed unpaired t-test with Welch's correction). Data are shown as mean ± SD. N refers to the number of biological replicates used in the experiment. Related to Supplementary Data 1. Source data are provided as a Source Data file. Created in BioRender. Sahu, A. (2026) https://BioRender.com/5q3eb5x.

support that reduced DUB activity in old mouse brains might derive from increased cysteine oxidation.

**DUB inhibition partially recapitulates age-related ubiquitylation**

After observing decreased DUB activity (this study) and increasing ubiquitylated protein levels[5] during brain aging, we next sought to understand the effect of global DUB inhibition on the ubiquitylated proteome. We acutely inhibited DUBs using a broad-spectrum DUB inhibitor, PR619[34], in 14-day post-differentiated human iPSC-derived neurons[35] (Fig. 3A). We chose this model as it has been utilized previously to investigate proteome-wide protein ubiquitylation[5,36,37] and to study human age-associated neurodegenerative diseases[38]. We chose a PR619 concentration and treatment duration that does not impart cellular toxicity (Fig. 3A, Fig. S3A) and minimizes off-target effects[39] while being sufficient to inhibit DUBs (Fig. S3B). Under these

conditions, PR619 treatment does not directly affect proteasome activity (Fig. S3C). In parallel, we also inhibited proteasome function using bortezomib as a control (Fig. 3A, Fig. S3A, B, C). We observed the expected increase in the total ubiquitylated protein levels (Fig. S3D). Subsequently, we enriched ubiquitylated peptides using a well-established antibody-based approach specific to the lysine di-GLY (K-Ɛ-GG) remnant motif[40] (Fig. 3A). Although this method leads to the enrichment of other modifications, such as ISGylation and NEDDylation, more than 95% of K-Ɛ-GG-modified sites are contributed by ubiquitylation[41]. Hence, from this point forward, we refer to K-Ɛ-GG-modified sites as ubiquitylated. In parallel, we measured proteome changes of treated-iNeurons using DIA-based MS (Fig.3A, Supplementary Data2). We noted a robust increase in the number of identified ubiquitylated peptides upon PR619 and bortezomib treatment (>14,000 Ub sites) compared to the DMSO control (>6000 Ub sites). In

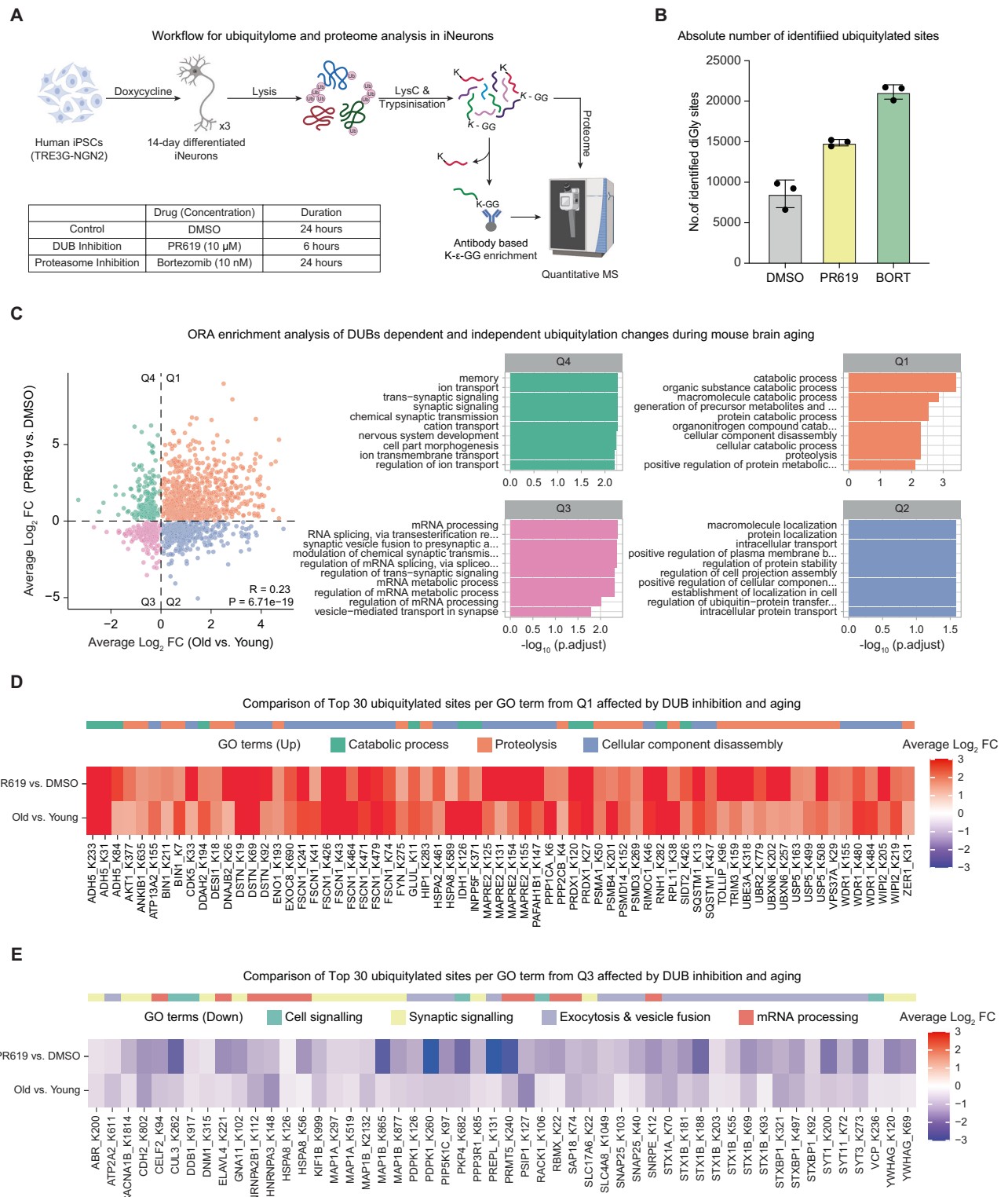

**Fig. 3 | DUB inhibition contributes to age-related ubiquitylation signatures.**
**A** Schematic illustrating the K-ε-GG antibody-mediated ubiquitylated peptide enrichment and total proteome analysis of iPSC-derived iNeurons treated either with DMSO, 10 μM PR619, or 10 nM Bortezomib (N = 3). **B** Number of identified ubiquitylated sites in iNeurons treated with the indicated drugs (N = 3; error bars represent the standard deviation from the mean). Data are shown as mean ± SD. **C** Left: Scatter plot comparing differentially enriched ubiquitylated sites between mouse aging (old vs. young; x-axis; from[5]) and DUB-inhibited iNeurons (PR619 vs. DMSO; y-axis; this study; N = 3). Right: Quadrant-based Over Representation Analysis (ORA) of the top 10 biological processes. Data includes ubiquitylated site

changes with adj.pvals <0.05 (for mouse) and Qvalue < 0.05 (for iNeurons). **D** Heatmap representing the comparison of average log2 FC of the top 30 enriched ubiquitylated sites per GO term from Quadrant 1 (Q1) in (**C**) between mouse aging and DUB-inhibited iNeurons. **E** Heatmap representing the comparison of average log2 FC of the top 30 enriched ubiquitylated sites per GO term from Quadrant 3 (Q3) in (**C**) between mouse aging and DUB-inhibited iNeurons. For both heatmaps, only differentially enriched ubiquitylated sites with adj.pvals <0.05 (for mouse) and Qvalue < 0.05 (for iNeurons) were used. N refers to the number of biological replicates used in the experiment. Related to Supplementary Data 2. Source data are provided as a Source Data file. Created in BioRender. Sahu, A. (2026) https://BioRender.com/5q3eb5x.

contrast, the number of total proteins quantified remained similar (~7000 proteins) (Fig.3B, Fig. S3E).

PCA based on global proteome data revealed that DUB inhibition by PR619 had a reduced impact on protein abundance compared to proteasome inhibition by bortezomib (Fig. S3F). On the other hand, both DUB and proteasome inhibition strongly affected the ubiquitylome compared to the vehicle control (Fig. S3G). Euclidean distance-based hierarchical clustering of 8481 cross-quantified ubiquitylated peptides revealed a more pronounced effect of bortezomib on ubiquitylated sites than PR619, as expected. Most of the commonly affected sites exhibited consistent changes (6251 sites showing increased and 824 decreased ubiquitylation, Fig. S3H, I). These data show that DUB inhibition leads to changes in protein ubiquitylation that are less pronounced but largely overlapping with proteasome inhibition. To gain a deeper understanding of the molecular function that might be impacted by DUB inhibition, we performed gene set enrichment analysis (GSEA) using proteins that showed altered ubiquitylation. We observed that proteins belonging to ligase activity, actin filament binding, cytoskeleton, and mRNA binding were among the top 10 terms associated with increased ubiquitylation following DUB inhibition (Fig. S3J).

To assess whether the ubiquitylated sites altered by DUB inhibition resemble those affected by aging, we compared significantly altered ubiquitylated sites in PR619-treated iNeurons with those observed in aging mouse brains[5] (Fig. 3C). Pearson correlation analysis revealed that DUB inhibition significantly and partially recapitulated aspects of age-induced ubiquitylated signatures (Pvalue = 4.4e-15, R = 0.23), suggesting that altered DUB activity may contribute to age-related molecular changes (Fig. 3C). To explore this overlap further, we analyzed GO terms enriched among ubiquitylated sites affected by both DUB inhibition and aging (Fig. 3C). Terms related to catabolic processes and proteolysis were among the top 10 enriched categories with increased ubiquitylation in both aging and following DUB inhibition. For example, increased ubiquitylation of proteins belonging to catabolic processes such as alcohol dehydrogenase 5 (ADH5), an enzyme involved in glutathione-dependent formaldehyde detoxification pathway and maintaining redox balance in cells[42] was observed under both aging and following DUB inhibition (Fig. 3D). Similarly, increased ubiquitylation of proteasome-associated subunits like PSMA1 and PSMB4, and autophagy-related proteins such as sequestosome-1 (SQSTM1) was detected across multiple sites (Fig. 3D). On the other hand, proteins involved in synaptic signaling, exocytosis and vesicle fusion, such as syntaxin-1B (STX1B) and its binding protein (STXBP1) showed reduced ubiquitylation during aging and following DUB inhibition (Fig. 3E). Together, these results demonstrate that the reduction of DUB activity may contribute to some of the age-associated ubiquitylation changes observed in the brains of aged mice.

Next, we asked whether we could relate ubiquitylation changes observed in the aging brain to reduced activity of a specific DUB. We selected Ubiquitin Specific Peptidase 7 (USP7) because its activity was reduced by at least 2.5-fold with age (Fig. 1F, Fig. S1F), and USP7 loss-of-function mutations have been causally associated with Hao−Fountain syndrome, a neurological disorder characterized by features of autism spectrum disorders[43]. To achieve partial inhibition of USP7 activity, 14-day post-differentiated iNeurons were treated for 24 h with the USP7-specific inhibitor P5091[44,45] under conditions that did not cause detectable cellular toxicity (Fig. S4A, B). Ubiquitylated peptides were subsequently enriched from treated iNeurons and analyzed by mass spectrometry alongside global proteome profiling (Fig. S4A, Supplementary Data 2).

Principal component analysis based on global proteome analysis and enriched ubiquitylated peptides revealed a clear separation between P5091-treated samples and DMSO controls (Fig. S4C, D). However, as with global DUB inhibition using PR619, P5091 treatment

exerted a markedly weaker effect on the proteome than on the ubiquitylome (Fig. S4E, F).

Although P5091 treatment did not significantly alter the abundance of previously reported USP7 substrates[19,46,47] under this treatment condition (Fig. S4E), altered ubiquitylation at one or more lysine residues was detected on multiple reported USP7 substrates, including UBE2E1, HUWE1, RNF220, NEDD4L, TRIP12, SIRT1, MAP4, MARCKS, and CCDC71L, indicating effective inhibition of USP7 activity by P5091 in iNeurons (Fig. S4F).

To determine whether USP7-dependent ubiquitylation changes resemble those occurring during aging, we compared significantly altered ubiquitylation sites in P5091-treated iNeurons with those identified in aging mouse brains (Fig. S4G). Pearson correlation analysis revealed a weaker association between aging- and USP7-induced changes (Fig. S4G), as compared to global DUB inhibition by PR619. However, a subset of proteins involved in macromolecule biosynthesis, nucleic acid metabolism, and their regulation displayed increased ubiquitylation both upon USP7 inhibition and during aging (Fig. S4G).

Further inspection of site-specific overlaps between aging and USP7 inhibition revealed increased ubiquitylation of proteins involved in the chaperone system (HSP90AA1, HSPH1, PPIA, SGTA), the ubiquitin-proteasome system (ATXN3, MINDY2, PSMB2, PSMD2, RAD23B, UBA1, UBXN1), cytoskeletal organization (ACTR1A, CORO2B, DCTN2, DCTN3, DYNC1H1, DYNLL1, MAP1B), and trafficking and sorting (AP2A1, CLTC, EPN1, EPN2) (Fig. S4H). In contrast, many proteins involved in synaptic transmission (SNAP25, STX1A, STX1B), ion-channel regulation (CACNA1B, CACNA1E, CACNG8, SLC17A6), and membrane trafficking exhibited decreased ubiquitylation upon both aging and USP7 inhibition (Fig. S4I).

Together, these data indicate that inhibition of USP7 in neurons can only partially recapitulate the aging-associated ubiquitylation changes observed in the brain. However, inhibition of USP7 is sufficient to alter the ubiquitylation of a subset of age-affected proteins involved in proteostasis, cytoskeletal dynamics, trafficking, and synaptic transmission.

## DUB decline precedes proteasome impairment in aging brains

Since both DUB and proteasome inhibition[5] can independently contribute to aging-like signatures, and emerging evidence suggests they may influence each other's function[48–52], we next investigated which of these molecular events occurs earlier during the aging process. To investigate this, we performed a temporal study analyzing DUB activity and proteasome activity, along with measuring the concentration of the reduced form of thiol (-SH) and total ubiquitylated protein levels in the brains of mice of different ages (Fig. 4A). We observed a stable thiol concentration until 12 months and then a significant decrease beginning from 18 months of age (Fig. 4B). In accordance with changes in thiol concentration, we observed a decline in DUB activity post-18 months of age compared to younger ages (Fig. 4C). On the other hand, we noticed a gradual accumulation of ubiquitylated proteins in the brain with age, mirroring the decrease in the DUB activity (Fig. 4D, Fig. S5A). Interestingly, we found a minor fluctuation in the activity of the proteasome across ages and significantly reduced activity emerging only after 24 months (Fig. 4E). A similar decline in DUB (Fig. S1E) and proteasome activity[3] has been observed in middle-aged killifish brains (12-13 wph). To assess whether changes in DUB and proteasome activity are associated with declining thiol concentrations during aging, we performed correlation analyses. We found that younger brains with higher thiol levels exhibit increased DUB and proteasome activity, while aged brains with lower thiol concentrations show reduced activity. Notably, the correlation with thiol concentration was stronger for DUB activity than for proteasome activity (Fig. S5B).

We next asked whether changes in DUB activity underlie the reduction in proteasome activity during aging. Earlier, we observed that acute inhibition of DUBs in iNeurons using 10 μM PR619 for 6 h did

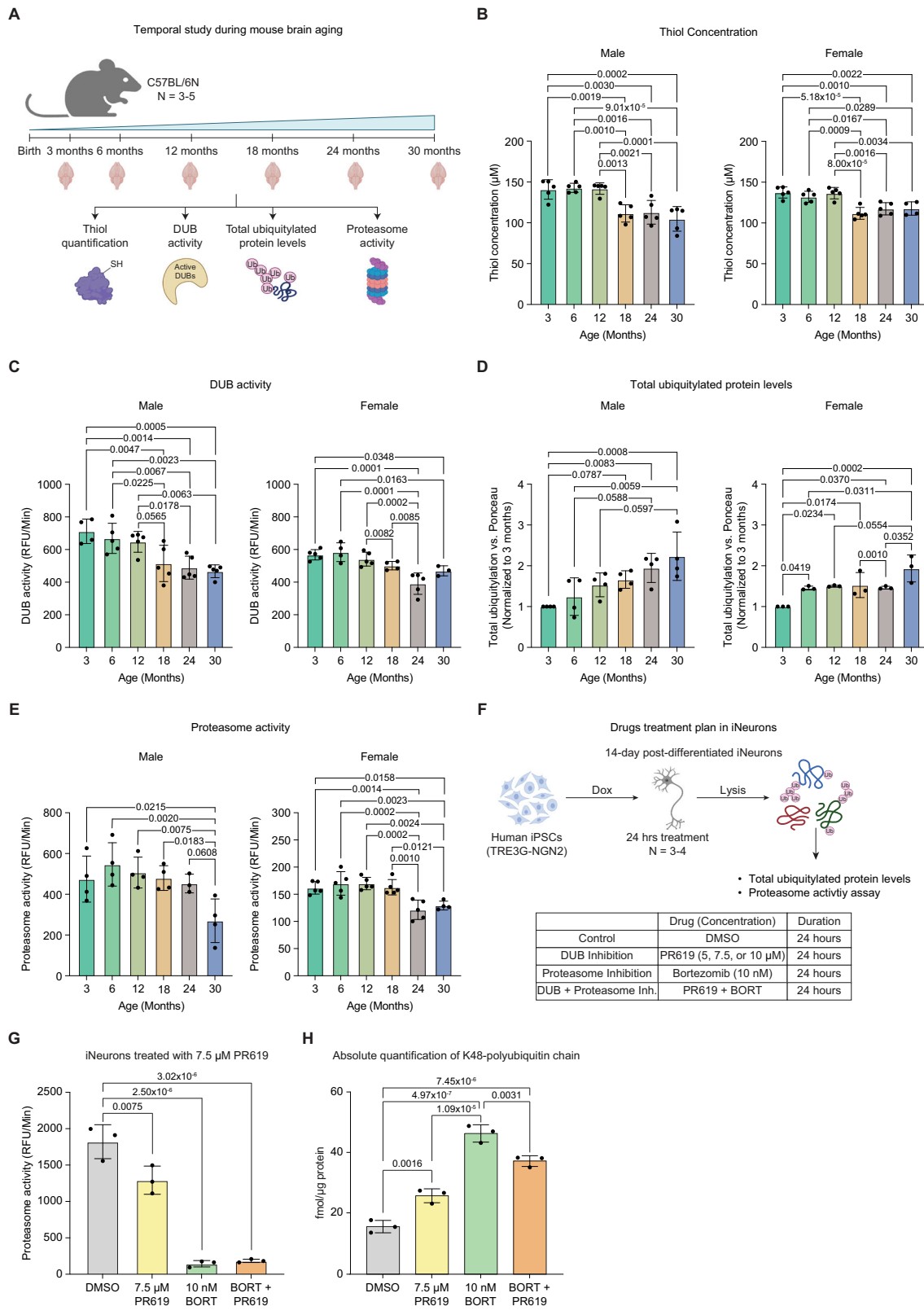

not alter proteasome activity (Fig. S3C). Thus, we hypothesized that chronic inhibition of DUBs could impact the proteasome. To test this, we inhibited DUBs in iNeurons using PR619 for 24 h at three different concentrations and bortezomib as a positive control (Fig. 4F). PR619 showed toxicity at 10 μM after 24 h, while 5 μM and 7.5 μM were non-toxic, even when combined with bortezomib. While 5 μM had no effect, 7.5 μM PR619 increased ubiquitylated proteins and significantly

reduced proteasome activity (Fig. 4G, Fig. S5C–E). To further assess proteasome-targeted ubiquitylation, we performed targeted pro-teomics using parallel reaction monitoring with absolutely quantified isotopically labeled spike-in reference peptides (AQUA-PRM). This analysis confirmed that 24-hour treatment with 7.5 μM PR619 resulted in increased levels of K48-linked polyubiquitin chains, the canonical signal for proteasomal degradation, although not to the same extent of

**Fig. 4 | Temporal study of molecular changes during mouse brain aging.**
**A** Schematic illustrating the temporal study of molecular changes during aging in C57BL/6N male and female mouse brains ($N$ = 3–5). **B** Reduced thiol concentrations in mouse brains across age groups ($N$ = 5 for all male and female groups, except $N$ = 4 for 30 months old females). **C** DUB activity across age groups ($N$ = 3 for 30 months old females; $N$ = 4 for 3 months old males and 6- and 18-months old females; $N$ = 5 for other groups). **D** Total ubiquitylated protein levels ($N$ = 3 for females; $N$ = 4 for males). **E** Proteasome activity ($N$ = 3 for 24 months old males; $N$ = 4 for other male groups and 30 months old females; $N$ = 5 for other female groups). **F** Schematic illustrating chronic DUB inhibition in iNeurons using

increasing concentrations of PR619 and 10 nM bortezomib for 24 h ($N$ = 3-4). **G** Proteasome activity in iNeurons treated with 7.5 µM PR619, 10 nM bortezomib, or both for 24 h ($N$ = 3). (**H**) Absolute quantification (AQUA-PRM) of K48-linked ubiquitin chains of iNeurons treated with 7.5 µM PR619, 10 nM bortezomib, or both for 24 h ($N$ = 3). One-way ANOVA was used for analysis in all panels. RFU Relative Fluorescence Units; Dox Doxycycline. Data are shown as mean ± SD in all panels. N refers to the number of biological replicates used in the experiment. Related to Supplementary Data 3. Source data are provided as a Source Data file. Created in BioRender. Sahu, A. (2026) https://BioRender.com/5q3eb5x.

bortezomib (Fig. 4H, Supplementary Data 3). Together, these results show that the decrease of DUB activity temporally precedes proteasome inhibition in the aging mouse brain, and that chronic impairment of DUBs in neurons leads to partial inhibition of proteasome activity.

### NACET treatment rescues age-associated molecular changes in aged brains

Building on our finding that DUB impairment (Fig. 1A, E) during brain aging is linked to oxidative stress and can be reversed (Fig. 2A, B), we tested whether antioxidant treatment could restore DUB function in aged animals. N-acetylcysteine ethyl ester (NACET) is a potent, orally bioavailable antioxidant that is known to cross the blood-brain and blood-retinal barriers. Once inside cells, NACET is rapidly de-esterified to N-acetylcysteine (NAC), which is then gradually deacetylated, providing a sustained supply of cysteine[53–55]. Cysteine acts directly as an antioxidant and can also promote activation of the NRF2 pathway. Moreover, as the rate-limiting precursor for GSH, cysteine supports the synthesis of this major cellular antioxidant. Because NACET increases intracellular cysteine and boosts GSH levels, with or without NRF2 activation[53–56], we hypothesized that NACET treatment could restore redox balance and recover DUB activity in aged brains. To test this, aged mice (22–24 months) were treated with NACET in drinking water for 12 days (Fig. 5A). NACET treatment significantly increased the pool of free thiols (Fig. 5B) and elevated NRF2 protein levels (Fig. S6A), but did not alter GSH levels (Fig. S6B) compared to vehicle controls in aged mouse brains. The increase in reduced thiol availability was accompanied by a restoration of DUB activity (Fig. 5C). Given that age-related DUB decline contributes to the accumulation of ubiquitylated proteins and proteasome dysfunction, we next assessed whether NACET could rescue these molecular changes. While total ubiquitylated protein levels showed a trend toward decreased accumulation, K48-linked polyubiquitylated protein levels were significantly reduced in NACET-treated brains (Fig. 5D, Fig. S6C). Consistently, proteasome activity was enhanced following NACET treatment in aged mice (Fig. 5E).

Since aging alters specific ubiquitin-chain linkage abundances in mouse brain[5], we next examined whether NACET could restore these molecular changes beyond proteostasis. Using targeted proteomics via parallel reaction monitoring with absolutely quantified isotopically labeled spike-in reference peptides (AQUA-PRM) (Fig. 5A), we found that NACET treatment decreased total ubiquitin and K48-linked ubiquitin levels (Fig. 5F, Supplementary Data 3), consistent with the immunoblot results. Additionally, NACET reduced the abundances of K11, K33, and K63 chains, which are known to increase during brain aging (Fig. S6D, Supplementary Data 3). Taken together, these findings demonstrate that NACET treatment restores redox homeostasis in aged brains, leading to the reactivation of DUB activity, a reduction in deleterious polyubiquitin chain accumulation, and an enhancement of proteasome function, thereby improving overall protein quality control during brain aging.

### Discussion

Aging is associated with a gradual decline in protein homeostasis, which contributes to the onset and progression of neurodegenerative

diseases. While proteasomal dysfunction has been extensively studied in the context of aging and age-associated pathologies[2,3,57–61], upstream regulators of ubiquitin turnover, such as deubiquitylating enzymes (DUBs), remain less explored. In this study, we systematically profiled the activity of cysteine protease DUBs during brain aging in both mouse and killifish, uncovering a redox-sensitive subset of enzymes that lose almost 40% of their catalytic activity with age despite unchanged protein abundance. Notably, several DUBs identified in our study, including ATXN3, UCHL1, UCHL5, and YOD1, have been previously implicated in neurodegenerative disorders such as spinocerebellar ataxia, Alzheimer's disease, and Parkinson's disease[12,22–24,62,63], underscoring the relevance of DUB activity loss in the context of aging and age-related neurodegeneration.

Mitochondrial dysfunction and impaired proteostasis, two hallmarks of aging and neurodegeneration, are tightly interconnected[1,29,64]. Mitochondria depend on proper protein import, folding, and degradation to function efficiently; failure in these processes can trigger mitochondrial dysfunction. In turn, dysfunctional mitochondria produce elevated levels of reactive oxygen species (ROS), which can oxidize amino acid residues such as cysteine and methionine. This leads to protein misfolding and impairs key enzymes involved in maintaining proteostasis, ultimately resulting in protein aggregation[65–68].

Our findings suggest that such ROS-driven thiol oxidation underlies the age-dependent loss of DUB activity in aged mouse brains, which was reversible upon in vitro treatment with dithiothreitol (DTT), a reducing agent. Furthermore, in vivo administration of the antioxidant N-Acetyl-L-cysteine ethyl ester (NACET) increased the pool of reduced thiols and restored DUB activity in aged animals without altering glutathione levels. These results are consistent with previous reports showing reversible oxidation and inactivation of cysteine protease DUBs, such as USP1, under oxidative conditions[13,69].

We confirmed the presence of multiple biomarkers indicative of redox imbalance in aged mouse brains, including reduced total thiol content, decreased NRF2 protein levels and GSH concentrations, as well as compensatory alterations in antioxidant enzyme expression, collectively consistent with elevated oxidative stress[70–73]. These data complement numerous previous studies that reported elevated ROS production in aged brains[74–76], and an age-dependent increase of metabolites such as cystine (oxidized cysteine), cysteine-glutathione disulfide, and 4-hydroperoxy-2-nonenal (4-HPNE), indicative of glutathione metabolism dysregulation during aging[71,77]. While redox imbalance likely represents a key contributing factor driving the loss of DUB function in the aging brain, a potential role of broader metabolic alterations and other post-translational modes of DUB modulation, such as phosphorylation, ubiquitylation, or protein-protein interactions, cannot be fully excluded[7].

Functionally, we demonstrate that inhibition of DUBs in human iPSC-derived neurons (iNeurons) recapitulates some of the ubiquitylation patterns observed in aged brains[5], establishing a causal link between DUB inactivation and age-related proteome changes. For example, alcohol dehydrogenase 5 (ADH5), one of the proteins showing the most substantial increase in ubiquitylation during aging and following DUB inhibition, is critical for formaldehyde detoxification via a glutathione-

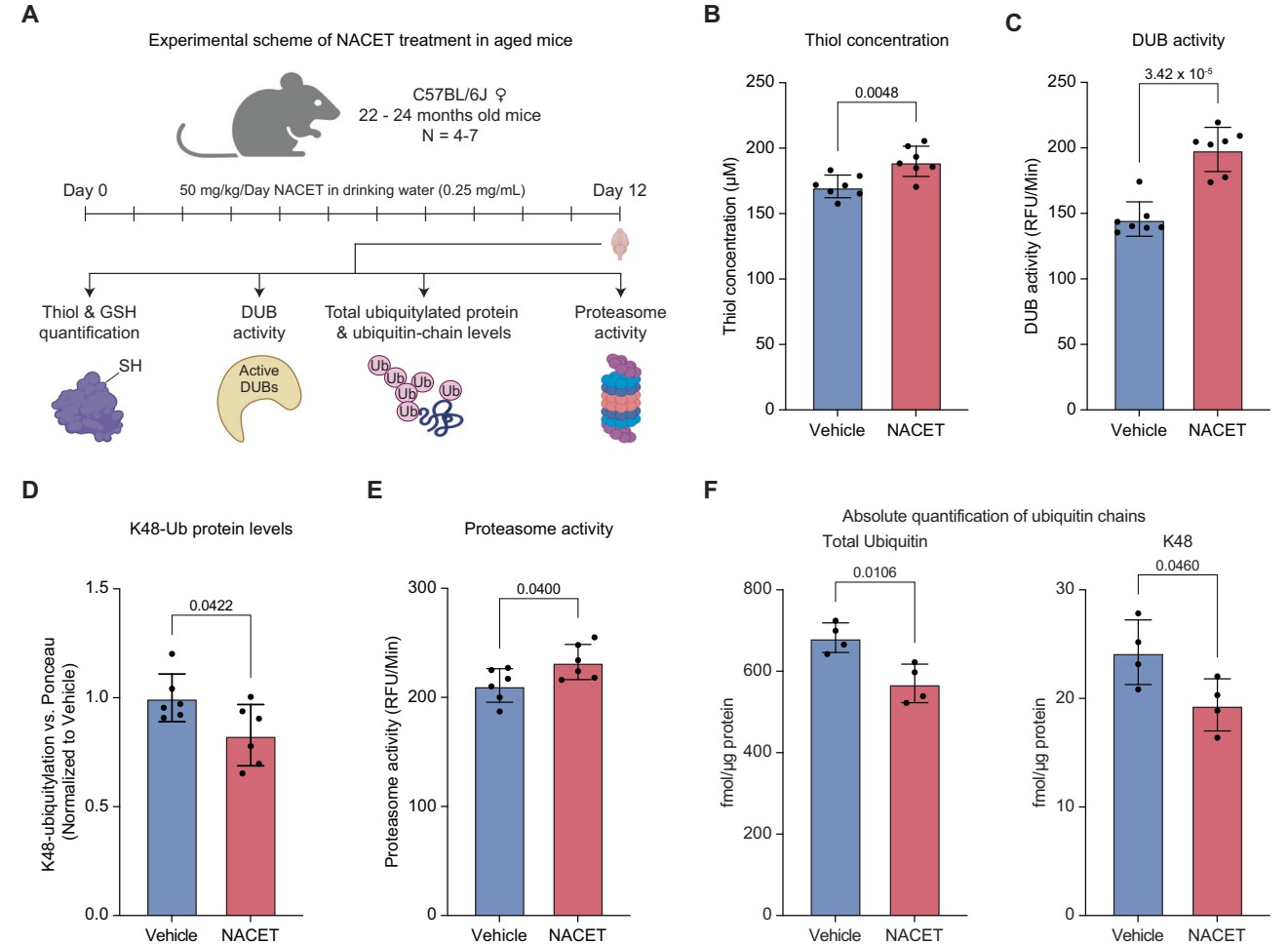

**Fig. 5 | NACET reverses age-associated alterations in redox balance and proteostasis. A** Schematic illustrating N-acetylcysteine ethyl ester (NACET) treatment in aged (22–24 months) C57BL/6J female mice. Animals were treated with 0.25 mg/mL NACET in drinking water to achieve a final concentration of 50 mg/kg/day of NACET (two independent cohorts; $N = 3$ and $N = 4$). **B** Reduced thiol concentrations in the NACET-treated mouse brains ($N = 7$). **C** DUB activity ($N = 7$). **D** K48 poly-ubiquitylated protein abundance quantification normalized to total protein levels ($N = 6$). **E** Proteasome activity ($N = 6$). Data in all panels are pooled from two independent experiments, with technical outliers excluded as described in the Methods section. **F** Absolute quantification (AQUA-PRM) of total ubiquitin and K48-linked ubiquitin chains in NACET vs. vehicle-treated aged mouse brains ($N = 4$). In all panels, two-tailed unpaired t-test with Welch's correction was used. RFU = Relative Fluorescence Units. Data are shown as mean ± SD in all panels. N refers to the number of biological replicates used in the experiment. Related to Supplementary Data 3. Source data are provided as a Source Data file. Created in BioRender. Sahu, A. (2026) https://BioRender.com/5q3eb5x.

dependent mechanism[78]. Impaired ADH5 function and formaldehyde toxicity have been linked to mitophagy, aging, and age-related neurological disorders[42,78,79], suggesting that its post-translational modification may affect protein stability or function.

Likewise, proteins involved in cytoskeletal dynamics, proteolysis, and protein folding exhibited increased ubiquitylation following DUB inhibition and in aged mouse brains[5]. Notably, Heat Shock 70 kDa Protein 8 (HSPA8), a chaperone whose reduced protein levels have been associated with the accumulation of neurodegeneration-related proteins and increased oxidative stress[80,81], shows increased ubiquitylation at most lysine residues during aging. However, specific sites such as K56 and K126 display reduced ubiquitylation in both aged brains[5] and Huntington's disease models[82]. Both K56 and K126 reside in the ATPase domain of HSPA8, and their post-translational modification has been implicated in modulating, particularly inhibiting chaperone activity[83-85]. The fact that these site-specific changes are reproduced by DUB inhibition but not by proteasome inhibition[5], highlights the distinct role of DUBs in maintaining HSPA8 chaperone activity by regulating its ubiquitylation.

Conversely, we also observed decreased ubiquitylation of a subset of proteins involved in synaptic function and mRNA processing upon

DUB inhibition in iNeurons. Proteins (such as Epsin 1 and Calcium/Calmodulin Dependent Protein Kinase II Alpha (CAMK2A), Supplementary Data 2) belonging to the same functional categories show reduced ubiquitylation levels during aging[5] and depolarized synaptosomes[86,87]. One possible explanation for such cases is an indirect regulatory effect, where disrupted DUB activity alters the coordinated interplay between DUBs and E3 ligases. For instance, inhibition of USP2a or USP7 has been shown to reduce p53 ubiquitylation by destabilizing the E3 ligase MDM2[88,89]. This highlights the importance of DUB-E3 interactions in shaping the ubiquitylated proteome landscape. A comprehensive analysis of how E1, E2, and E3 ligase activity is modulated during aging and how this contributes to age-associated proteome remodeling and disease remains an important future direction. Tools such as Ub-Dha-based activity probes could be instrumental in mapping these dynamics[90].

Our data indicate a widespread, age-dependent decline in deubiquitylating enzyme activity affecting multiple DUBs. Consistent with this, we demonstrate that partial inhibition of a single DUB, namely USP7, is sufficient to alter the ubiquitylation of a subset of age-sensitive proteins involved in proteostasis, intracellular trafficking, cytoskeletal organization, and synaptic signaling, yet only weakly recapitulates the

global ubiquitylation landscape associated with aging. This limited overlap, particularly when contrasted with broad DUB inhibition, supports the conclusion that age-associated proteome remodeling arises from the coordinated functional decline of multiple cysteine DUBs rather than from the dysfunction of a single enzyme.

Building on our findings of decreased global DUB activity (this study) and proteasome decline during aging[3,5], we next sought to determine the sequence of molecular events leading to proteostasis disruption in the aging brain. A temporal analysis across different mouse age groups revealed a progressive, thiol oxidation-mediated reduction in DUB activity from middle to old age (12–18 months). In contrast, the proteasome's chymotrypsin-like (CT-L) activity varied with age and showed a significant decline only in later life, consistent with observations of 20S proteasome fluctuations during aging reported by ref. [91]. Furthermore, in line with findings from[92] that reported reduced proteasome activity in SH-SY5Y dopaminergic cells following USP14 knockout, and[52], that reported impaired proteasome function in OLN-t40 oligodendroglial cells upon global DUB inhibition by PR619, we observed diminished CT-L proteasome activity following chronic (24 h) DUB inhibition in iNeurons. Collectively, these results suggest that DUB dysfunction is an early molecular event in the proteostasis cascade, preceding and potentially contributing to proteasome impairment during aging. Although the CT-L assay used here reflects the combined activity of multiple proteasome species, coordinated declines in both 20S and 26S proteasome activity have been reported in aging brains[93–95]. Nonetheless, we acknowledge that age-related compensatory mechanisms, such as an increased 20S/26S activity ratio or elevated immunoproteasome function, may mask or offset a more pronounced decline in 26S proteasome activity[3,91].

Consistent with a model of chronic, redox-driven DUB inactivation during physiological aging, we further demonstrate that NACET treatment not only restored DUB activity but also rescued proteasome function in aged mouse brains. This effect may be explained by at least three potential mechanisms: (1) enhanced DUB activity reverses stress-induced ubiquitylation on proteasome subunits such as RPN10 or RPN13, which otherwise promotes dissociation of the 19S regulatory particle from the 26S complex. This autoinhibitory mechanism is thought to have evolved as a protective strategy to prevent the engagement of ubiquitylated substrates with defective or stalled proteasomes, thereby minimizing futile degradation attempts under stress conditions[96–98]; (2) by relieving oxidative stress, NACET may independently promote 26S complex reassembly and enhance ATPase activity, thereby restoring proteasome function[99–102]; or (3) both mechanisms may act in parallel to achieve the observed rescue effect. As a broad-spectrum antioxidant and direct ROS scavenger, NACET may also preserve the reduced state of catalytic cysteines in E3 ligases and other redox-sensitive enzymes, thereby indirectly reducing protein misfolding, ubiquitylation burden, and proteasomal stress.

While these findings highlight the potential of redox-based interventions in late life to restore aspects of proteostasis and demonstrate the reversibility of oxidative inactivation, several important limitations remain. NACET treatment in this study was limited to 12 days, and the long-term safety, efficacy, and durability of redox-based interventions in aging brains remain to be determined. Notably, prolonged NACET administration at comparable doses was well tolerated in young animals[55], supporting its feasibility; however, whether similar outcomes extend to aged brains requires further investigation. Moreover, although we establish a link between redox-driven DUB inactivation and proteostasis decline, the relationship between altered DUB function and other canonical hallmarks of brain aging, as well as downstream physiological and behavioral consequences, remains to be systematically explored.

Our findings contrast with a body of literature reporting that genetic or pharmacological inhibition of proteasome-associated DUBs, such as USP14 and UCHL5, can enhance proteasome activity across various systems, including in vitro assays, cultured cells, and *C. elegans* models[49,50,103]. Similarly, inhibition of USP30 has demonstrated neuroprotective effects in Parkinson's disease models[104,105], while USP7 inhibition has been shown to reduce neuroinflammation and enhance clearance of amyotrophic lateral sclerosis (ALS)–related proteotoxic proteins[106,107]. Numerous other DUBs have also been implicated in neurodegenerative contexts[108,109], as comprehensively reviewed by ref. [28]. Importantly, the reported increase in proteasome activity and enhanced clearance of toxic proteins upon DUB inhibition appears to be highly context-dependent. These effects are most consistently observed in *C. elegans*, cancer cell lines, or disease models characterized by abnormal protein accumulation. Such benefits may be mediated through mechanisms including 20S proteasome gate opening, reduced ubiquitin chain trimming, or non-catalytic regulatory effects of DUBs on proteasome function[51].

In conclusion, we identified a subset of DUBs that undergo selective and reversible inactivation with age due to thiol oxidation. This loss of deubiquitylation capacity emerges as an early molecular feature of aging that precedes proteasomal decline and contributes to proteostasis disruption. Our findings provide mechanistic insight into the hierarchical dysfunction of the ubiquitin system in the aging brain and suggest novel opportunities to restore proteostasis by preserving cysteine proteases DUB function.

## Methods

### Mice

Wild-type male and female mice (*Mus musculus*) were either C57BL/6J or C57BL/6N, obtained from Janvier Labs or internal breeding at FLI. Animals were kept in a specific pathogen-free animal facility with unlimited access to food and water at FLI, Jena (C57BL/6J) or ZMG, Halle (C57BL/6N). The housing conditions were adjusted to a 12-hour day-night cycle, temperature and humidity of 20 °C ± 2 °C and 55% ± 15, respectively. The following aged mice were used in the experiment and are mentioned in each figure: 3, 6, 12, 18, 22, 24, and 30 months old. In all figures, N refers to the number of biological replicates used. C57BL/6J mice were euthanized with $CO_2$, whereas C57BL/6N mice were sacrificed by cervical dislocation (approved code: K2bM3). Brains were isolated, washed in PBS, cut into two halves, weighed, immediately snap-frozen in liquid nitrogen or dry ice, and stored at −80 °C. The guidelines from the European Parliament on the protection of animals, Directive 2010/63/EU, and the guide for the care and use of laboratory animals, 8th edition, 2011, Washington (DC), were used for all experiments. Sacrifice and organ collection were carried out following §4(3) of the German Animal Welfare Act.

### Killifish

All experiments were conducted using the male *Nothobranchius furzeri* strain MZM-0410 ($N = 3$ biological replicates) in accordance with institutional and national guidelines. Fish were bred and maintained in the FLI fish facility in accordance with §11 of the German Welfare Act. Sacrifice and organ collection were carried out following §4(3) of the German Animal Welfare Act. Fish were housed individually or in groups (maximum one fish per 1.7 L) in recirculating systems (Aqua Schwarz, Göttingen, Germany) with centralized filtration, maintained under a 12-hour day-night cycle. Water temperature (27 °C ± 1 °C), conductivity (-2.5 mS), and pH were continuously monitored and recorded. As environmental enrichment, group-housed fish were provided with certified contaminant-screened blue polycarbonate igloos and shelters (Bio-Serv, Flemington). Newly hatched larvae were fed twice daily with live *Artemia nauplii* and weaned onto live bloodworms (*Chironomidae*) once daily from 4 to 6 weeks of age. Fish health was assessed daily by trained caretakers using the FLI killifish health score sheet[110], and animals were euthanized by rapid chilling if humane endpoints were reached. Health monitoring followed the protocols described in ref. [110].

## NACET treatment in aged mice

For N-acetylcysteine ethyl ester (NACET, C7H13NO3S, Advanced ChemBloks, O32426, Lot #AC99952A) treatments on aged mice, 22 to 24 months old female C57BL/6J mice were used. Following established protocols[53,54], NACET was resuspended in drinking water at 0.25 mg/mL. Then, based on the average daily water intake of ~5 mL per mouse, 50 mg/kg of NACET was administered daily to each animal, and a freshly prepared NACET solution was replaced every 2 days. Control mice received regular drinking water (vehicle). After 12 days of treatment, mice were sacrificed, and brains were collected as described above and stored at −80 °C for downstream analyses. The experiment was performed independently twice, with $N = 3$ and $N = 4$ biological replicates per experiment. Animal experiments were approved by the Thuringian State Office for Consumer Protection (Thüringer Landesamt für Verbraucherschutz, license number: FLI-24-011).

## Activity-based DUB probe labeling in brain tissues

Mouse and killifish brains were lysed in DUB buffer consisting of 50 mM Tris-HCl (pH 7.5), 150 mM NaCl, 0.1% NP-40, 5% CHAPS, 5 mM MgCl$_2$, 5 mM β-mercaptoethanol, and 10% glycerol. Killifish brains were lysed using a vial tweeter (Amplitude 100%, cycle 0.9), while mouse brains were manually disrupted by mechanical trituration. This was performed by repeatedly aspirating and dispensing the tissue using a 2 mL syringe (Injekt) fitted first with a 0.70 × 30 mm needle, followed by a 0.45 × 25 mm needle (Sterican). After lysis, samples were subjected to a short centrifugation step to pellet debris, and the supernatants were transferred to fresh 1.5 mL centrifuge tubes. Protein concentrations were determined by measuring absorbance at 280 nm using a NanoDrop 2000 spectrophotometer (Thermo Scientific). Mouse brain lysates equivalent to 230 μg of protein were either reduced with 10 mM dithiothreitol (DTT) or mock-treated with DUB buffer (control) for 30 min at 300 rpm and 25 °C. Both mouse and killifish lysates were subsequently alkylated with 10 mM N-ethylmaleimide (NEM; Thermo, Cat# 23030) for 30 min at 300 rpm and 25 °C to block free sulfhydryl groups, serving as a negative control for activity-based probe labeling.

Following NEM or DMSO treatment, samples were incubated with a 2.3 μM mixture of three activity-based probes: Biotin-Ahx-Ub-VME (UbiQ-054), Biotin-Ahx-Ub-PA (UbiQ-076), and Biotin-Ahx-Ub-VS (UbiQ-188), for 60 minutes at 300 rpm and 25 °C. The reaction was terminated by adding 0.4% SDS, followed by freezing and storage at −20 °C. The volume corresponding to ~30 μg of protein was taken out from each reaction. Of this, 10 μg was used for immunoblotting to validate probe labeling, and 20 μg was stored at −20 °C for future use. Excess unbound probe was removed from the remaining reaction mixture (~200 μg protein) using 10 kDa Amicon Ultra centrifugal filters (Merck, UFC5010) with DUB buffer containing 2% SDS.

## Enrichment and digestion of probe-labeled DUBs

To enrich DUBs labeled by the activity-based probes, an automated workflow using the Agilent Bravo AssayMAP platform was employed, with minor modifications based on[111]. Streptavidin cartridges were first equilibrated with 200 μL of PBS at 10 μL/min and acetylated with 50 μL of 10 mM sulfo-NHS acetate. Before sample loading, the cartridges were equilibrated with 300 μL of DUB buffer (internal cartridge wash 1) at 20 μL/min. Filtered lysates were loaded at 10 μL/min. After binding, the cartridges were washed once with 250 μL of DUB buffer containing 2% SDS and twice with 250 μL of 50 mM ammonium bicarbonate (AmBic) at 10 μL/min. Proteins were digested on-cartridge using 0.5 μg of LysC (Cell Signaling) in 30 μL of 50 mM AmBic. Digestion was performed in a 6 μL reaction volume at 45 °C for 60 minutes without a reaction chase. Peptides were eluted in two steps using 100 μL of 50 mM AmBic (no internal cup wash, 10 μL/min). Eluates were subsequently digested with trypsin (0.5 μg/μL, Promega) at 37 °C and 500 rpm overnight. Reactions were acidified with 10% trifluoroacetic acid (TFA) to a pH < 3. Peptides were desalted using Waters Oasis® HLB μElution Plate (30 μM) according to the manufacturer's instructions and sent for DUB identification and quantification using LC-MS/MS, as mentioned below.

## Mouse brain homogenization

Snap-frozen half-brains stored at −80 °C were thawed on ice and transferred to a 5 mL Douncer. The volume of pre-chilled PBS added to each brain sample was calculated based on the estimated protein content (~5% of the fresh tissue weight) to reach a 20 μg/μL concentration. Homogenization was performed for 15–20 passes of the pestles up and down the glass cylindrical Douncer, followed by 15 s of spin. The homogenization step was repeated for three cycles, and the total homogenate was transferred to a 1.5 mL centrifuge tube. The homogenates were clarified from tissue debris by centrifuging at 17,000 × g for 1 min at 4 °C, and the clarified homogenate was aliquoted to multiple 1.5 mL centrifuge tubes. The aliquots were then stored at −80 °C for future experiments.

## DUB activity assay

To 5 μL of brain homogenates, 75 μL 1× DUB assay buffer (without DTT), supplied with Abcam's Deubiquitinase Assay Kit (ab241002), was added. The lysates were prepared by sonication for 60 sec ON/ 30 sec OFF at 4 °C for five cycles in a Bioruptor Plus sonicator at high intensity setting. The lysates were centrifuged at 10,000 × g for 5 min at 4 °C, and the supernatants were then transferred to a new 1.5 mL centrifuge tube. The protein concentration was measured at A280 using Thermo Scientific NanoDrop 2000. Lysates corresponding to 5 μg of protein were used to measure DUB activity as per the manufacturer's protocol. As a negative control, NEM was used at a final concentration of 10 mM per reaction. Fluorescence was measured after setting up the reaction in the kinetic mode for 60 minutes at 25 °C by TECAN kinetic analysis (excitation 350 nM, emission 440 nM, 30-second reading interval) on a Safire II microplate reader (TECAN). The DUB activity was determined as the difference between the activity of protein lysates and the residual activity of the lysate in the presence of NEM, and results were represented as relative fluorescence units per minute (RFU/Min). Similar steps were followed for the iNeurons lysate treated with DMSO, PR619, or Bortezomib, except that RFU was calculated from the difference between the activity of protein lysates in the presence and absence of DUB substrates. Outliers were removed based on raw fluorescence data, if pipetting errors or air bubbles were identified after measurement, or if the final RFU value fell outside one standard deviation from the mean.

## Proteasome chymotrypsin-like activity assay

To 7 μL brain homogenates, 50 μL 1× lysis buffer consisting of a final concentration of 1× Proteasome assay buffer (supplied with the UBP-Bio's Proteasome Activity Fluorometric Assay Kit II, J4120), 50 mM NaCl, 2 mM ATP, 5 mM MgCl2, and 10% Glycerol. While for the iNeurons pellet, 100 μL 1× lysis buffer was used. The lysates were prepared, and protein concentration was measured as mentioned in the DUB activity assay methodology. The final concentration of 1× Suc-LLVY-AMC fluorogenic peptides supplied with the kit was used to measure the chymotrypsin-like activity (CT-L) of the proteasome using 50 μg of proteins in a total of 100 μL reaction as per the manufacturer's protocol using a 96-well plate (Falcon, 353219). This assay reflects the combined activity of multiple proteasome species, including 20S, 26S, and immunoproteasome complexes, rather than selectively measuring 26S activity. Fluorescence was measured after setting up the reaction in the kinetic mode for 60 minutes at 37 °C by TECAN kinetic analysis (excitation 360 nM, emission 460 nM, 30-second reading interval) on a Safire II microplate reader (TECAN). The CT-L activity was determined as the difference between the activity of protein lysates and the residual activity of the lysate in the presence of 100 μM MG132 supplied

with the kit, and results were represented as Relative Fluorescence Units per minute (RFU/Min). Similar steps were followed for the iNeurons lysate treated with DMSO, PR619, or Bortezomib, except that RFU was calculated from the difference between the activity of protein lysates in the presence and absence of fluorogenic peptides. Outliers were removed based on raw fluorescence data, if pipetting errors or air bubbles were identified after measurement, or if the final RFU value fell outside one standard deviation from the mean.

## Thiol concentration measurement using the DTNB assay

To 15 μL of brain homogenate, 50 μL of reaction buffer was added. The reaction buffer contained 0.1 M sodium phosphate dibasic (Sigma, S0876) and 1 mM EDTA (Roth, 8040.3) at pH 8.0. The lysate preparation and protein concentration measurement were conducted as outlined in the DUB activity assay methodology. For the experiment, lysates corresponding to 100 μg of proteins were combined with 5 μL of 5,5′-dithio-bis-[2-nitrobenzoic acid] (DTNB, Thermo Scientific, 22582) to achieve a total volume of 280 μL in a 96-well plate (Falcon, 353219). L-Cysteine (Sigma, C7352) standards were prepared at concentrations of 0, 10, 50, 100, 200, 400, 800, and 1000 μM in the reaction buffer, and set up the reaction as experimental samples using DTNB. The reactions were then incubated for 15 min at 22 °C. The absorbance was measured at 412 nm using the Safire II microplate reader (TECAN). The experimental samples were corrected using non-treated lysates as blanks for measuring thiol concentration. Based on the absorbance vs. cysteine concentration trendline curve equation derived from the L-cysteine standards, the thiol concentration for the unknown experimental samples was extrapolated. Outliers were removed based on raw fluorescence data, if pipetting errors or air bubbles were identified after measurement, or if the final thiol concentration values fell outside one standard deviation from the mean.

## Immunoblot analysis

Samples for immunoblot analysis were used as prepared and mentioned previously. Otherwise, mouse brain homogenate or iNeurons pellets were lysed in 1× lysis buffer containing 2% SDS and 25 mM HEPES for 60 sec ON/ 30 sec OFF at 20 °C for ten cycles in a Bioruptor Plus sonicator at high-intensity setting. Lysates were boiled at 95 °C for 5 minutes, followed by centrifugation for 1 minute at 17,000 × g. Supernatants were transferred to new 1.5 mL centrifuge tubes, and protein concentrations were measured using Pierce BCA Protein Assay Kit (Thermo Scientific, 23225). Subsequently, samples were reduced using DTT (Carl Roth, 6908) at a final concentration of 10 mM for 15 min at 45 °C. 10–20 μg of proteins were used along with 4x sample loading buffer containing 1.5 M Tris pH 6.8, 20% (w/v) SDS, 85% (v/v) glycerin, 5% (v/v) β-mercaptoethanol. The sample mixture was then incubated at 95 °C for 5 minutes. For immunoblots, proteins were separated on 4–20% Mini-Protean® TGX™ Gels (Bio-Rad #4561096) by sodium dodecyl sulfate-polyacrylamide gel electrophoresis (SDS-PAGE) using a Mini-Protean® Tetra Cell system (Bio-Rad, Neuberg, Germany, 1658005EDU). For NRF2, Bis-Tris 4–12% gradient MES buffer gel was used. Proteins were then transferred using a Trans-Blot® Turbo™ Transfer Starter System (Bio-Rad #1704150) onto nitrocellulose membrane (Carl Roth, 200H.1). For measurement of total protein levels on blot, membranes were stained with Ponceau S (Sigma, P7170-1L) for 5 minutes on a shaker (Heidolph Duomax 1030), followed by washing with Milli-Q water. The Ponceau staining was then imaged on a Molecular Imager ChemiDocTM XRS+Imaging system (Bio-Rad) and destained by three washes in TBST (Tris-buffered saline (TBS, 25 mM Tris, 75 mM NaCl), with 0.5% (v/v) Tween-20) for 10 minutes each. The blots were then incubated in PBS for 1 hour in a house-made blocking buffer of 3% BSA (w/v) and 0.5% Tween 20 (v/v). Post-blocking blots were incubated overnight at 4 °C on a tube roller (BioCote® Stuart® SRT6) with primary antibodies against total ubiquitin P4D1 (1:1000, Santa Cruz #sc8017), Lys-48 specific anti-ubiquitin antibody (1:1000,

Sigma Aldrich #05-1307), NRF2 (1:2000, Proteintech #16396-1-Ap), Tubulin (1:5000, Proteintech #66031-1-Ig) or Streptavidin-HRP (1:20,000, Abcam #ab7403) in an enzyme dilution buffer containing 0.2% BSA (w/v) and 0.1% Tween20 (v/v) in PBS. The next day, blots were washed three times with TBST for 10 minutes each at room temperature, and Streptavidin-HRP blots were imaged while ubiquitin blots were incubated with horseradish peroxidase-coupled secondary antibody (Dako #P0447 or #P0448) at room temperature for 1 h (1:2000 in 0.3% (w/v) BSA in TBST). Following incubation with secondary antibody, blots were washed 3 times for 10 min in TBST. The chemiluminescent signals were detected using an ECL Pierce detection kit (Thermo Fisher Scientific, Waltham, MA, USA, 32109) on the Molecular Imager ChemiDocTM XRS + Imaging system (Bio-Rad). The results were analyzed using Bio-Rad's Image Lab 6.1 software. For protein normalization, Ponceau was used.

## Human iPSC maintenance

The WTC11 human induced pluripotent stem cells (iPSCs) were maintained according to[35] and were a kind gift from the Ward Lab at the National Institute of Health, US. Briefly, the tissue culture surface was coated with 1× Matrigel (Corning, 356231) coating solution prepared in DMEM/F12 (Gibco, 11320033) medium for at least 1 h and then removed from the surface. iPSCs were thawed from liquid nitrogen and washed once with DMEM/F12, followed by resuspended in Essential 8 (E8) culture medium (Gibco, A1517001) supplemented with Y-27632 ROCK inhibitor (Abcam, ab120129) (Day 1). The cells were incubated at 37 °C. The culture was maintained by daily E8 medium change without the presence of 10 μM ROCK inhibitor. Every fourth day (Day 4), cells were split by detaching using Accutase (Sigma, A6964), resuspended in E8 medium supplemented with ROCK inhibitor, and seeded into a new Matrigel-coated surface. For storage in liquid nitrogen, Day 4 cells were frozen in cryopreservation medium containing 20% FBS and 10% DMSO in E8 medium.

## iPSC differentiation to iNeurons and drug treatment regimen

Once iPSCs reached 70–80% confluency (Day 4 of iPSC), they were detached using Accutase, pelleted down at 300 × g for 5 min. The cell pellets were then resuspended and transferred to a Matrigel-coated surface in induction medium (IM) (DMEM/F12 (Gibco, 11320033) supplemented with N2 supplement (Gibco, 17502048), L-glutamine (Gibco, 25030081), MEM NEAA (Gibco, 11140050), and 2 μg/mL doxycycline (Sigma, D9891)) supplemented with 10 μM Y-27632 ROCK inhibitor (Abcam, ab120129). The cells were incubated at 37 °C (Day 1 of iNeurons). The next day, the culture was maintained by changing the IM medium without ROCK inhibitor (Day 2). On the same day, new plates were coated with 1 mg/mL Poly-L-Ornithine (PLO, Sigma Aldrich, P4957) in the buffer containing 100 mM boric acid (Roth, 6943.1), 75 mM sodium chloride (Roth, P029.2), 25 mM sodium tetraborate (Sigma, 221732), and 1 M sodium hydroxide (Day 2). A day after (Day 3), the PLO-coating was removed from the surface, washed three times with PBS, and dried under the hood for at least 30 minutes. Meanwhile, differentiated cells were dissociated from the Matrigel-coated plate's surface using Accutase, washed with DMEM/F12, and pelleted down. The cells were then resuspended in cortical neuron culture medium (CM) consisting of Neurobasal medium (Gibco, 10888022) supplemented with B-27 supplement (Gibco, 17504044), 10 ng/mL NT-3 (PeproTech, 450-03), 10 ng/mL BDNF (PeproTech, 450-02), 1 μg/mL laminin (Sigma-Aldrich, L2020), 2 μg/mL doxycycline (Sigma, D9891), and 10 μM Y-27632 ROCK inhibitor and seeded to the PLO-coated surface. The iNeurons were then cultured by removing half of the old medium and exchanging it with fresh CM medium without ROCK inhibitor biweekly. For long-term storage in liquid nitrogen, Day 3 differentiated iNeurons were resuspended in 20% FBS and 10% DMSO in CM media and aliquoted as 6 million/ mL per vial.

The DUB and proteasome inhibition experiments were performed on 14-day post-differentiated iNeurons. PR619 (Sigma, 662141) and P5091 (MCE, HY-15667)[44,45] were used to inhibit cysteine DUBs and USP7 specifically, while the proteasome was inhibited using Bortezomib (Sigma, 5043140001). Drugs were prepared in DMSO and treated to iNeurons by removing half of the old medium and exchanging it with fresh CM medium with double the required final drug concentrations. In all experiments, the final concentration of Bortezomib was 10 nM, and the cells were treated for 24 h. 10 µM PR619 for 6 h was used to inhibit DUBs in the ubiquitylated peptides enrichment experiment. 5, 7.5, and 10 µM PR619 for 24 h were used to assess the impact of DUB inhibition on the proteasome. 2.5 µM P5091 for 24 h was used to partially inhibit USP7 specifically. For cell harvesting, iNeurons were washed three times with chilled PBS. Accutase was added to the cells and incubated for 10 min at 37 °C. After 10 min, the reaction was stopped by adding DMEM/F12. Using a cell scraper, iNeurons were scraped gently from the surface, collected into 1.5 mL centrifuge tubes, and pelleted down at 1000 × g for 1 minute. Pellets were then stored at −80 °C for downstream experiments.

## Sample preparation for proteome and ubiquitylated peptides analysis

Brain homogenates corresponding to 1.1 mg and iNeurons pellets were lysed in 1× lysis buffer containing 2% SDS and 25 mM HEPES final concentrations for 60 sec ON/ 30 sec OFF at 20 °C for ten cycles in a Bioruptor Plus sonicator at high-intensity setting. Lysates were boiled at 95 °C for 5 min, followed by centrifugation for 1 minute at 17,000 × g. Supernatants were transferred to new 1.5 mL centrifuge tubes, and protein concentrations were measured using Pierce BCA Protein Assay Kit (Thermo Scientific, 23225). Subsequently, samples were reduced using DTT (Carl Roth, 6908) at a final concentration of 10 mM for 15 min at 45 °C. Proteins corresponding to 1.1 mg were then alkylated using freshly prepared iodoacetamide (IAA) (Sigma-Aldrich, I1149) with a final concentration of 15 mM for 30 min at room temperature in the dark. Cold acetone corresponding to 4 times the volume was added to the samples and incubated overnight at −20 °C, as described in ref. 112. The next day, acetone was removed post-centrifugation at 17,000 × g at 4 °C, washed twice with 80% ice-cold acetone, and resuspended in digestion buffer consisting of 3 M urea and 100 mM HEPES at pH 8.0 such that the final protein concentration was 1 µg/µL. Subsequently, proteins were digested using 1:100 (enzyme: protein) LysC (Wako sequencing grade, 125-05061) at 37 °C for 4 hours, followed by dilution with HPLC water to make a 1.5 M final urea concentration and digestion with 1:100 Trypsin (Promega sequencing grade, V5111) for 16 hours. The next day, the reaction was stopped by acidifying peptides with 10% trifluoroacetic acid (v/v). Peptides corresponding to 20 µg and 1000 µg were taken for proteome and ubiquitylated peptides enrichment, respectively. Peptides were desalted using Waters Oasis® HLB µElution Plate (30 µM, 2 mg for proteome and 30 mg for ubiquitylome) following the manufacturer's instructions. The cleaned peptides were then dried using a vacuum concentrator at 45 °C and reconstituted in 5% acetonitrile (v/v) and 0.1% formic acid (v/v) MS Buffer A. For total proteome analysis, 20 µg of diluted peptides at 1 µg/µL were transferred to MS vials, spiked with iRT kit peptides (Biognosys, Ki-3002-2), and sent for LC-MS/MS. Meanwhile, samples were further processed for ubiquitylated peptide enrichment, as described below.

## Enrichment of ubiquitylated peptides

Dried peptides corresponding to ~1000 µg were used to enrich ubiquitylated peptides. The PTMScan® HS Ubiquitin/SUMO Remnant Motif (K-ε-GG) kit (Cell Signaling Technology, 59322) was used, and peptides were enriched according to the manufacturer's instructions. Thereafter, the enriched K-ε-GG modified peptides were desalted,

concentrated, and prepared in MS vials for the LC-MS/MS analysis as described above.

## Absolute quantification of ubiquitin chain linkages

Followed the standardized protocol for parallel reaction monitoring (PRM)-based measurement of endogenous ubiquitin chain linkages using Absolute QUAntification (AQUA) synthetic peptides, as described by us before[5], with minor modifications. Briefly, following acetone precipitation and resuspension in digestion buffer, protein samples corresponding to 20 µg in 20 µL were spiked with AQUA peptides into each sample at a concentration of 20 fmol per 1 µg of estimated protein content prior to digestion. Subsequent processing was performed as described in the "Sample preparation for total proteome and analysis of ubiquitylated peptides" section, with the only modification being the use of 10% formic acid instead of 10% trifluoroacetic acid (TFA) for peptide acidification prior to desalting. Peptides were desalted using a Waters Oasis® HLB µElution Plate (30 µm, 2 mg) and resuspended in 10 µL of MS Buffer A. Peptide concentration was re-assessed using a Thermo Scientific NanoDrop 2000 at A280. Peptides were transferred to MS vials, and 3 µL of the peptide mixture was injected into a nanoAcquity UPLC M-Class system coupled to an Orbitrap Fusion Lumos mass spectrometer, following the protocol described in ref. 5. Peak group identification was carried out using SpectroDive (12.0.24) and subsequently verified manually. Quantification employed a spike-in strategy, calculating the ratio between endogenous (light) and reference (heavy) peptides to achieve absolute quantification. All AQUA peptides corresponding to total ubiquitin and various linkage types were quantified, with the exception of K29.

## Data-independent acquisition mass spectrometry

For DUB analysis on the Evosep platform, ubiquitylome and proteome of P5091-treated iNeurons, and proteome of NACET- and vehicle-treated mouse brains, samples were loaded onto Evotips following the manufacturer's instructions and using protocols adapted from[111]. Briefly, peptides were separated using the Evosep One system (Evosep, Odense, Denmark) equipped with either an 8 cm × 150 µm i.d. column packed with 1.5 µm Reprosil-Pur C18 beads (Evosep Performance, EV-1109, PepSep, for mouse DUB profiling, ubiquitylome and proteome of iNeurons, and NACET- and vehicle-treated mouse brains proteome) or a 15 cm × 150 µm i.d. column packed with 1.9 µm Reprosil-Pur C18 beads (Evosep Endurance, EV-1106, PepSep, for killifish DUB profiling) for a 44-minute gradient (30 samples per day, 30SPD). Solvent A consisted of water with 0.1% formic acid, and solvent B was acetonitrile with 0.1% formic acid. Liquid chromatography was coupled to an Orbitrap Exploris 480 mass spectrometry (Thermo Fisher Scientific). A detailed description of the mass spectrometry parameters used for data-independent acquisition (DIA) analysis is available in ref. 111.

For proteome and ubiquitylome analysis of PR619-treated iNeurons, peptides were separated in trap/elute mode using a nanoAcquity MClass UPLC system (Waters Corporation, Milford, MA, USA) equipped with a trapping column (Waters nanoEase M/Z Symmetry C18, 5 µm, 180 µm × 20 mm) and an analytical column (Waters nanoEase M/Z C18 HSS T3, 1.7 µm, 75 µm × 250 mm). Solvent A consisted of water with 0.1% formic acid, and Solvent B was acetonitrile with 0.1% formic acid. Approximately 1 µL of sample (~1 µg on-column) was loaded onto the trapping column at 5 µL/min using Solvent A. After a 6-minute trapping step, peptides were eluted through the analytical column at 0.3 µL/min. During the elution step, the gradient of Solvent B increased nonlinearly from 0% to 40% over 120 minutes. The total LC run time was 145 minutes, including column equilibration and conditioning. The LC was interfaced with an Orbitrap Exploris 480 mass spectrometry (Thermo Fisher Scientific, Bremen, Germany) via a Proxeon nanospray source using a Pico-Tip emitter (360 µm OD × 20 µm ID, 10 µm tip; New Objective). The spray voltage was set to 2.2 kV, with the capillary and ion

transfer tube temperatures both maintained at 300 °C. The RF ion funnel was set to 30%. For data-independent acquisition (DIA), full MS scans were acquired over an m/z range of 350–1650 at a resolution of 120,000 (FWHM), with a maximum injection time of 60 ms and a $3 \times 10^6$ ions AGC target. The default precursor charge state was set to $3^+$. DIA MS/MS scans used 40 variable-width isolation windows across the MS1 range, with stepped normalized collision energies of 25, 27.5, and 30%. MS/MS spectra were acquired at a resolution of 30,000 with a fixed first mass of 200 m/z, using either 35 ms maximum injection time or until $3 \times 10^6$ ions were collected. All data were acquired in profile mode using Xcalibur 4.3 and Tune version 2.0 (Thermo).

### Analysis of intracellular glutathione (GSH)

To measure total intracellular GSH levels, whole brains were homogenized in ice-cold 4% (w/v) Trichloroacetic Acid (TCA) containing 1 mM Tripotassium Ethylenediaminetetraacetate ($K_3$EDTA, 1 mL per 100 mg tissue). Samples were processed immediately. Free thiols were derivatized with the fluorescent probe monobromobimane (mBrB; Calbiochem, La Jolla, CA, USA) as previously described[113]. Quantification was performed using an Agilent Series 1100 HPLC system (Agilent Technologies, Milan, Italy) equipped with diode array and fluorescence detectors.

### Data processing for mass spectrometry DIA data

Spectral libraries for DUB profiling in mouse and killifish brains, and ubiquitylated peptide and proteome analyses in iNeurons were generated by searching the DIA run using Spectronaut Pulsar (Biognosys, Zurich, Switzerland), as referenced in Supplementary Table 1. All searches were performed against species-specific protein databases supplemented with a list of common contaminants. Search parameters for variable modifications included Oxidation (M) and Acetyl (protein N-term) across all datasets. In addition, Biotin was included for DUB profiling, while GlyGly (K) was used for ubiquitylome datasets as part of variable modifications. For both ubiquitylome and proteome analysis, carbamidomethylation (C) was used as a fixed modification. Three missed cleavages were allowed for ubiquitylome, whereas two were allowed for the rest of the datasets. Library-based searches were performed with a 1% FDR (false discovery rate) at both the protein and peptide levels. Relative quantification was performed in Spectronaut using the LFQ QUANT 2.0 method with global normalization, precursor filtering percentile utilizing a fraction of 0.2, and global imputation. No imputation was applied for DIA total proteome analysis in iNeurons, and local normalization was used. Relative quantification between conditions was performed using replicate samples and default settings in Spectronaut. Candidate and report tables were exported for downstream analyses using R studio (4.2).

### Compilation of cysteine DUBs and their interacting partners

Cysteine protease deubiquitylases (DUBs) were compiled using the Mouse Genome Informatics (MGI) database (GO:0101005) from the Jackson Laboratory and the DUBase repository[19]. Known DUB interactors were annotated based on data from[18] and DUBase. When necessary, protein entries were mapped to their human gene name orthologs prior to inclusion or use in downstream analyses.

#### GO enrichment analysis

Gene Set Enrichment Analysis (GSEA) was performed using the gseGO function from the clusterProfiler R package[114]. Protein entries were first mapped to their human gene name orthologs and used as input for the enrichment analysis.

### Reporting summary

Further information on research design is available in the Nature Portfolio Reporting Summary linked to this article.

### Data availability

The proteomics data generated in this study have been deposited in the MassIVE repository with the following identifiers: DUB activity profiling of killifish brains: MSV000098114, DUB activity profiling of mouse brains: MSV000098115 (cohort 1) and MSV000100403 (cohort 2), Ubiquitylome and proteome of PR619-treated iNeurons: MSV000098116, Ubiquitylome and proteome of P5091-treated iNeurons: MSV000100405, AQUA-PRM analysis of ubiquitin chains in PR619-treated iNeurons: MSV000100404, and AQUA-PRM analysis of ubiquitin chains in NACET-treated animals: MSV000098802. Source data are provided with this paper.

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

## Acknowledgements

The authors gratefully acknowledge the support of the FLI Proteomics Core Facility (Emilio Cirri, Norman Rahnis, Therese Dau, and Nadine Pömpner), the Mouse and Fish Facilities, and the ZMG Animal Facility. We also thank the Rudolph group at FLI for sharing mice and reagents. We are grateful to Paulius Grigaravicius and Omid Omrani (FLI), as well as Matthias Ebert (ZMG), Doreen Sander, Jennifer Kopietz, and Pascal Rudewig, for their assistance with organ isolation. A.O. is supported by the Else Kröner-Fresenius-Stiftung (award number: 2019_A79), the Fritz Thyssen Foundation (award number: 10.20.1.022MN), the Chan Zuckerberg Initiative Neurodegeneration Challenge Network (award numbers: 2020-221617, 2021-230967, and 2022-250618), and the NCL Stiftung. A.K.S., A.M., P.R.W., A.S., T.P., and A.O. are supported by the German Research Foundation (Deutsche Forschungsgemeinschaft, DFG) through the Research Training Group ProMoAge (GRK 2155) and T.P. was additionally supported by DFG grant 566709590. A.S. was funded by the European Union and the State of Saxony Anhalt as part of the Thera4Age project (project ID ZS/2024/01/183688). F.G. gratefully acknowledges support from the Fondo di Beneficenza Intesa Sanpaolo (project ID: B/2023/0171). F.N. was supported by the AIRC foundation (MFAG 2021 ID 26038). The FLI is a member of the Leibniz Association and is jointly funded by the Federal Government of Germany and the State of Thuringia. We thank Ilona Renken-Olthoff and the entire team of the Health and Medical University for their support. Some of the figures were created using BioRender. Grammarly and ChatGPT were used for language editing.

## Author contributions

Conceptualization: A.K.S., T.P., and A.O. Data curation: A.K.S., A.l.M., D.D.F., and A.M. Investigation: A.K.S., A.l.M., D.D.F., A.M., P.R.W., D.G., C.G., and R.R. Methodology: A.K.S., A.l.M., and D.D.F. Project administration: T.P. and A.O. Data analysis: A.K.S. Supervision: F.N., F.G., A.S., T.P., and A.O. Visualization: A.K.S. Writing-original draft: A.K.S. and A.O. Writing-review & editing: A.l.M., D.D.F., A.M., and T.P.

## Funding

## Competing interests

The authors declare no competing interests.
