## [Transparent Peer Review file · Nature Communications]

Oxidative stress causes a reversible decrease of deubiquitylases activity in old vertebrate brains

Corresponding Author: Dr Alessandro Ori

Version 0:

Reviewer comments:

Reviewer #1

(Remarks to the Author)

The ubiquitin-proteasome system is a key determinant of aging and proteostasis, but the role of DUBs in this process is less well understood. Given their critical function in deubiquitinating proteins, studying how DUBs change under stress conditions and during aging is of particular importance. In this study, the authors combined multiple approaches and models (killifish, mouse and iPSC-derived neurons) to gain insights into DUB activity in neuronal proteostasis. To this end, they profiled cysteine protease DUBs using state-of-the-art activity-based proteomics. They found that a redox-sensitive subset of DUBs loses activity during aging, even when protein levels remain unchanged. Additionally, they showed that oxidative stress impairs DUB function via thiol oxidation, and that antioxidant treatment can restore DUB activity both in vitro and in vivo. Finally, they demonstrate that DUB inhibition in iPSC-derived neurons recapitulates changes observed in aged killifish brains. These results can have important implications for understanding the aging process. The conclusions are supported by solid experimental approaches. Nevertheless, I have several comments that I hope the authors will address.

Major points

- The figure legends should include clear statistical information, including indication of significance levels.
- In the Figure 1c, the PCA analysis shows notable variation for NEM-treated samples in old mice. I recommend discussing the possible reasons and implications of this variation in the manuscript.
- The authors conclude that: "The decrease of DUB activity is largely independent of protein abundance (Fig. S1A), suggesting the involvement of post-translational mechanisms influencing DUB activity in old brains". The conclusion is stated after the killifish data, but Fig. S1A corresponds to mouse data. It would be important to make this more clear in the text. Alternatively, the authors could consider including a similar volcano plot comparing total protein levels in young and old killifish using available datasets.
- Indeed, Fig. S1A (total proteomics) shows that the protein levels of many DUBs do not change with age. However, in Fig. S1A, it appears that the authors used a proteomics dataset detecting ~35 DUBs. Since there are ~90 DUBs identified in mouse, this dataset covers less than half of them. Moreover, the number of DUBs in the total proteomics dataset is also lower than in the activity-based pulldown assays. An important point to clarify is how many of the DUBs detected in Fig. S1A are cysteine protease DUBs, and whether they were identified in both the total proteomics and the activity-based pulldown assays. To strengthen this section, the authors could consider adding a graph or table listing the age-dysregulated DUBs at the activity level alongside their corresponding total protein levels (or indicating if they were not detected in the total proteomics dataset). This would make the interpretation clearer. Also, the authors should indicate in the text that the DUB activity assays and proteomics for protein abundance were performed on mice at slightly different ages (30 weeks for activity, 33 weeks for proteomics).
- In the case that the most age-dysregulated DUBs at the activity level are not detected in the total proteomics dataset, the authors could consider to assess by Western blot whether their total protein levels of key DUBs remain unchanged during (if there are good antibodies available).

- The DUB inhibition experiments were performed using the broad DUB inhibitor PR619. There are specific inhibitors for specific DUBs such as USP7, one of the cysteine protease DUBs whose activity decreases with age. For key experiments presented in Figs. 3 and 4, the authors could consider using specific inhibitors targeting the most dysregulated cysteine protease DUBs during aging, if such inhibitors are available. These experiments would help determine whether specific age-dysregulated DUBs drive the majority of age-related changes in ubiquitination and proteasome activity. Even if the decline in activity of these DUBs only accounts for a subset of age-related ubiquitination changes, the results would still be valuable for identifying potential DUB targets with implications in aging.

- Figure 5: As an antioxidant, NAC can have multiple intracellular effects and influence various signaling pathways and regulators beyond cysteine protease DUB activity. Therefore, NAC could affect ubiquitination levels and proteasome activity through several mechanisms, making it challenging to establish a direct link between DUB activity, proteasome activity, and aging. The authors should address this in more detail in the discussion.

Minor points

- In line 44, please add "(proteostasis)" after "protein homeostasis."

- Lines 47-49: The group of Ulrich Hartl showed that proteasome activity increases with age in *C. elegans* (Walther et al., Cell 2015, Figure S2c). Thus, a decline in proteasome activity during aging might not happen in all animals. Nevertheless, I agree with the authors that discussing this in detail in the introduction may be not necessary. Instead, the authors could consider rephrasing the sentence as follows: "Alterations in proteasome activity and the ubiquitylome have been well characterized in the context of aging in both invertebrates and vertebrates."

Reviewer #2

(Remarks to the Author)

Reviewer #3

(Remarks to the Author)

The study by Sahu and colleagues finds that Deubiquitinating enzyme function declines with age in the brains of mice and killifish, that this is driven not by changes in DUB abundance but oxidative state and that ex vivo treatment with antioxidants can rescue these changes. The findings are interesting, the work is technically sophisticated and the claims are impactful. All of which makes the work of interest with an overall moderate impact. However, there are a number of findings in where the claims made by the researchers are not fully supported by their data and additional experiments would be needed to validate these claims.

1) The number of animals is Fig 1 / Fig S1 is very low $N = 3$ and only males. If possible, it would be important for researchers to show that key findings are replicable in a larger cohort and not just the product of chance variation from a small cohort size.

2) The Researchers claim that age-related changes in DUB activity cannot be fully explained by changes in protein abundance but do not provide a satisfactory demonstration of this (only showing a volcano plot where some DUBs are significantly reduced while others significantly increase with age). If anything, the data that there is significant DUB remodeling with age and would support changes in abundance of DUBs as a driver of these changes.

3) Fig S1A it would be important to label names of the DUBs marked in red so readers can compare their change in abundance with the change in activity of these DUBs with age shown in Fig S1C. The researchers should also normalize the prevalence of active DUBS in Fig S1C to the relative abundance of these DUBS in Fig S1A to show that the changes they report are indeed not driven by changes in abundance.

4) The researchers claim the following factors are enriched but these changes are all non-significant PSMD7, PSMD14, and COPS5 (Fig 1d $P=0.15$ to $P=0.66$)

5) The researcher claim that declines in DUB activity precede impaired proteasome function but this seems to be based just on chymotrypsin-like activity. This measure is going to record not just 26S activity (which is relevant to the researchers question) but also 20S (whose levels some researchers find increases with age in response to diminished 26S) and also immunoproteasome which may increase with age driven by inflammatory signalling. As such this claim is potentially misleading the researchers should do native PAGE activity blots or similar to look at the activity of 26S in their samples.

Reviewer #4

(Remarks to the Author)

This manuscript investigates the regulatory mechanisms of deubiquitinating enzymes (DUBs) activity in the context of brain

aging in vertebrates. It establishes that DUBs activity significantly declines with age in two vertebrate models, independent of protein abundance. The study elucidates that oxidative stress-mediated thiol oxidation is a pivotal mechanism underlying the decline in DUBs activity, a process that is reversible. Furthermore, it reveals that the reduction in DUB activity occurs prior to the impairment of proteasome function, thereby providing new evidence for the early molecular events associated with protein homeostasis imbalance during brain aging. The manuscript also validates the rescue effect of redox regulation on molecular disorders related to brain aging through interventions using NACET.

However, the current version of the manuscript contains several shortcomings that require attention.

Major :

1. The manuscript's mixed use of experimental animal strains and genders may introduce systematic errors: DUB activity assays were performed using male C57BL/6J mice (line-130), male C57BL/6N mice were used in Figure 4, and female C57BL/6J mice were used for NACET treatment (Figure 5). However, no evidence was provided to indicate: 1) the basis for the consistency in brain aging-related molecular characteristics (such as ROS levels and DUB activity) between C57BL/6J and C57BL/6N; 2) whether the pattern of DUB activity changes exhibits sex differences, and whether the rescuing effect of NACET is applicable only to females.
2. The results of DUB activity confirmed that "the activity of UCHL1 and YOD1 decreased, while the activity of CYLD and USP9X remained unchanged" (Fig S1D), but only four examples of DUBs were presented, and no enrichment analysis results of all differentially active DUBs were provided. This weakens the specificity of the conclusion that cysteine protease DUBs are sensitive to oxidative stress.
3. Oxidative stress was indirectly suggested through "changes in antioxidant enzyme expression" (Supplementary Figure S2A) and "decreased thiol concentration" (Figure 2A), but without directly detecting markers of direct oxidative damage such as ROS levels, MDA, or protein carbonylation levels in the aged brain, the possibility that thiol oxidation is caused by metabolic disorders rather than ROS cannot be ruled out.
4. NACET treatment only validated the rescue at the molecular level, and it was unable to prove that the molecular-level improvements could translate into the restoration of brain function; furthermore, the GSH levels in the brain after NACET treatment were not detected, making it impossible to confirm whether "thiol restoration" was achieved through GSH-mediated redox balance reconstruction.
5. The discussion did not address the key limitations: 1) NACET treatment lasted only 12 days, and the safety and efficacy of long-term intervention remain unknown; 2) the association between changes in DUBs activity and other aging markers was not explored.

Minor :

1. The molecular weights of the bands in the Western blots results of image S4D were not labeled.
2. PR619, NACET, DUB probes, etc., require the addition of supplier and product numbers to facilitate the replication of related experiments by subsequent researchers.

Version 1:

Reviewer comments:

Reviewer #1

(Remarks to the Author)

The authors have addressed all my comments and I support publication of the manuscript.

Reviewer #2

(Remarks to the Author)

Reviewer #3

(Remarks to the Author)

The Authors did an excellent job addressing all of my comments prior comments.

Reviewer #4

(Remarks to the Author)

All concerns raised by the reviewers have been addressed and resolved by the authors, with no additional issues requiring response.

REVIEWER COMMENTS

Reviewer #1 (Remarks to the Author):

The ubiquitin-proteasome system is a key determinant of aging and proteostasis, but the role of DUBs in this process is less well understood. Given their critical function in deubiquitinating proteins, studying how DUBs change under stress conditions and during aging is of particular importance. In this study, the authors combined multiple approaches and models (killifish, mouse, and iPSC-derived neurons) to gain insights into DUB activity in neuronal proteostasis. To this end, they profiled cysteine protease DUBs using state-of-the-art activity-based proteomics. They found that a redox-sensitive subset of DUBs loses activity during aging, even when protein levels remain unchanged. Additionally, they showed that oxidative stress impairs DUB function via thiol oxidation, and that antioxidant treatment can restore DUB activity both in vitro and in vivo. Finally, they demonstrate that DUB inhibition in iPSC-derived neurons recapitulates changes observed in aged killifish brains. These results can have important implications for understanding the aging process. The conclusions are supported by solid experimental approaches. Nevertheless, I have several comments that I hope the authors will address.

Major points:

1. The figure legends should include clear statistical information, including indication of significance levels.

We thank the reviewer for this valuable comment. In all figure panels, statistical significance is indicated numerically (Pvalues) rather than with asterisks, to facilitate precise interpretation. The corresponding figure legends specify the statistical tests used to assess significance levels for each comparison.

2. In the Figure 1c, the PCA analysis shows notable variation for NEM-treated samples in old mice. I recommend discussing the possible reasons and implications of this variation in the manuscript.

We thank the reviewer for this insightful observation. The increased variability observed in the old NEM-treated brain samples is most likely attributable to technical factors associated with the treatment and sample processing. To address concerns regarding robustness, we have now included an independent

replication of the whole experiment (cohort 2, N = 3 biological replicates). As suggested, we have also discussed this variability in the manuscript lines 98-99, where it is stated: “We analyzed a first cohort of samples comprising brains from 3 young (3 months old) and 3 old (33 months old) C57BL/6J male mice.”

We hope these additions adequately address the reviewer’s concern.

3. The authors conclude that: “The decrease of DUB activity is largely independent of protein abundance (Fig. S1A), suggesting the involvement of post-translational mechanisms influencing DUB activity in old brains”. The conclusion is stated after the killifish data, but Fig. S1A corresponds to mouse data. It would be important to make this more clear in the text. Alternatively, the authors could consider including a similar volcano plot comparing total protein levels in young and old killifish using available datasets.

We apologize for the confusion and appreciate the reviewer's prompt attention to this matter. The statement referring to Fig. S1A (volcano plot of DUB abundance) indeed originated from the mouse dataset. This reference was first introduced at lines 86-88 (stated earlier as “We observed a decrease in the global DUB activity in old (30 months) mouse brain lysates compared to young (3 months) ones, which cannot be explained by changes in DUB protein abundance (proteomics data from 5)”), immediately following the mouse DUB activity results using biochemical assay, and was later restated after the killifish data, which may have caused confusion.

To improve clarity and ensure a more balanced cross-species comparison, we have removed the volcano plot of DUB protein abundance in the mouse brain (previously Fig. S1A) and instead included comparative heatmaps depicting both DUB activity and protein abundance in mouse and killifish brains during aging (now presented as Fig. 1F and Fig. S1F). Additionally, we have incorporated clarifying statements into the manuscript (lines 132–140).

For the reviewer’s convenience, we have attached the updated heatmaps and the corresponding revised text below. Please note that the figure order and layout have been slightly adjusted to improve the logical flow and clarity of presentation.

“Importantly, in both species, the age-dependent decline in DUB activity was largely independent of corresponding changes in DUB protein abundance (Fig. 1F, Fig. S1F). In mouse brains, 20 out of 27 DUBs that exhibited age-dependent changes in activity in at least one cohort showed no significant alteration in protein abundance (absolute \log_2 FC > 0.58 and Pvalue < 0.05) (Fig. 1F). Similarly, in aging killifish brains, 5 out of 6 DUBs displaying reduced activity did not show changes in protein abundance, with the exception of USP25, which

exhibited decreased protein levels (Fig. S1F). For a subset of DUBs whose total protein abundance was not detected (7 in mouse and 1 in killifish), it was not possible to assess whether the observed activity changes were independent of protein abundance.”

DUBs enrichment vs. abundance during aging in mice

Figure 1F: Heatmap comparing age-associated changes in DUB activity (this study; cohort 1: 30 vs. 3 months; cohort 2: 33 vs. 3 months; N = 3) and protein abundance (TMT proteomics from (Marino et al. 2025); 33 vs. 3 months). Values represent Pvalues from unpaired t-tests with Welch’s correction. ‘NA’ indicates DUBs not detected. DUBs highlighted in red exhibit significant (Pvalue < 0.05) age-associated activity changes in at least one cohort.

DUBs enrichment vs. abundance during aging in killifish

Figure S1F: Heatmap comparing age-associated changes in DUB activity (this study; 39 vs. 5 wph; N = 3) and protein abundance (DIA proteomics from (Di Fraia et al. 2025)) in killifish brains. Values represent Pvalues from unpaired t-tests with Welch’s correction. ‘NA’ indicates DUBs not detected. DUBs highlighted in red exhibit significant (Pvalue < 0.05) age-associated activity changes.

- Indeed, Fig. S1A (total proteomics) shows that the protein levels of many DUBs do not change with age. However, in Fig. S1A, it appears that the authors used a proteomics dataset detecting ~35 DUBs. Since there are ~90 DUBs identified in mouse, this dataset covers less than half of them. Moreover, the number of DUBs in the total proteomics dataset is also lower than in the activity-based pulldown assays. An important point to clarify is how many of the DUBs detected in Fig. S1A are cysteine protease DUBs, and whether they were identified in both the total proteomics and the activity-based pulldown assays. To strengthen this section, the authors could consider adding a graph or table listing the age-dysregulated DUBs at the activity level alongside their corresponding total protein levels (or indicating if they were not detected in the total proteomics dataset). This would make the interpretation clearer. Also, the authors should indicate in the text that the DUB activity assays and proteomics for protein abundance were performed on mice at slightly different ages (30 weeks for activity, 33 weeks for proteomics).

We thank the reviewer for this detailed and constructive comment, which helped us clarify the dataset scope and improve the presentation.

In the DUB activity-based enrichment assays, we detected 56 and 54 cysteine DUBs in cohorts 1 and 2, respectively. In comparison, our TMT-based total proteomics dataset for mouse (Marino et al. 2025) identified 39 cysteine DUBs. Among the 27 DUBs that showed altered activity in at least one cohort (highlighted in red in the heatmap), corresponding abundance measurements were available for 20 DUBs in the TMT dataset. The remaining seven DUBs (BAP1, MINDY4, OTUD1, OTUD5, USP2, USP21, and USP28) were not detected in the total proteomics dataset; thus, for these proteins, we cannot determine whether activity changes were independent of protein abundance.

To facilitate a more direct comparison of activity and abundance, we have now included new heatmaps (Fig. 1F and Fig. S1F; also referenced in our earlier responses) that display changes in DUB activity alongside corresponding protein abundance for both mouse and killifish brains. DUBs not detected in the proteomics dataset are marked as “NA.” This visualization enables readers to readily assess which DUBs exhibit age-dependent activity changes relative to their protein levels. Accordingly, we have incorporated the clarification into the Results section in lines 132–140.

Finally, we have clarified the ages of the animals used for each assay in the figure legends, noting that the DUB activity assays were performed on mice at 30 weeks, whereas the TMT proteomics data were generated from 33-week-old mice. We believe these additions substantially improve the clarity and interpretability of this section.

5. In the case that the most age-dysregulated DUBs at the activity level are not detected in the total proteomics dataset, the authors could consider to assess by Western blot whether their total protein levels of key DUBs remain unchanged during (if there are good antibodies available).

We thank the reviewer for this thoughtful suggestion. We would like to clarify that our total proteomics dataset already provides protein abundance information for 20 out of the 27 DUBs that exhibited age-dependent changes at the activity level, including key enzymes such as ATXN3, OTUB2, OTUD7A, UCHL1, and USP7. For the remaining seven DUBs, BAP1, MINDY4, OTUD1, OTUD5, USP2, USP21, and USP28, protein levels were not detected in our total proteomics dataset, nor are they reported in publicly available mouse brain aging proteome resources. Similarly, in aging killifish brains, 5 out of 6 DUBs displaying reduced activity did not show changes in protein abundance, with the exception of USP25, which exhibited decreased protein levels. Therefore, for these specific DUBs, we are unable to assess whether their age-associated activity changes occur independently of total protein abundance. We commented about this in lines

138–140, which now reads as, “For a subset of DUBs whose total protein abundance was not detected (7 in mouse and 1 in killifish), it was not possible to assess whether the observed activity changes were independent of protein abundance.”

6. The DUB inhibition experiments were performed using the broad DUB inhibitor PR619. There are specific inhibitors for specific DUBs such as USP7, one of the cysteine protease DUBs whose activity decreases with age. For key experiments presented in Figs. 3 and 4, the authors could consider using specific inhibitors targeting the most dysregulated cysteine protease DUBs during aging, if such inhibitors are available. These experiments would help determine whether specific age-dysregulated DUBs drive the majority of age-related changes in ubiquitination and proteasome activity. Even if the decline in activity of these DUBs only accounts for a subset of age-related ubiquitination changes, the results would still be valuable for identifying potential DUB targets with implications in aging.

We thank the reviewer for this valuable suggestion to further dissect the contribution of individual DUBs to age-related changes in ubiquitylation and proteasome function. Based on our activity profiling, we selected USP7 as a candidate, as its activity was reduced by at least 2.5-fold with age (Fig. 1F and Fig. S1F). In addition, loss-of-function mutations in USP7 have been causally linked to Hao–Fountain syndrome, a neurological disorder characterized by features of autism spectrum disorders, underscoring the relevance of USP7 to neuronal function.

In line with the reviewer’s recommendation, we therefore performed targeted inhibition of USP7 using the selective small-molecule inhibitor P5091 (Chauhan et al. 2012; Weinstock et al. 2012). Fourteen-day post-differentiation iNeurons were treated with P5091 (2.5 μ M for 24 hours) under non-toxic conditions, followed by enrichment of ubiquitylated peptides and quantitative proteomics. These new experiments and their results have now been included in the Results section under the heading “DUB inhibition partially recapitulates age-related ubiquitylation signatures” (lines 339–380), with the corresponding data presented in Fig. S4.

We note that proteasome activity was not directly measured in the P5091-treated neurons. However, previous studies have reported that P5091 primarily inhibits USP7 catalytic activity without globally perturbing proteasome function (Chauhan et al. 2012), particularly under short-term and partial inhibition conditions. Consistent with this, we did not observe widespread changes in global protein abundance, supporting the interpretation that the observed effects predominantly

reflect alterations in ubiquitylation rather than secondary consequences of altered proteasomal degradation.

Overall, our results demonstrate that selective inhibition of USP7 is sufficient to reproduce a subset of aging-associated ubiquitylation changes of proteins involved in macromolecule biosynthesis, proteostasis, cytoskeletal organization, trafficking, and synaptic function, although to a markedly lesser extent than global DUB inhibition with PR619. This supports the conclusion that age-related remodeling of the neuronal ubiquitylome likely arises from the combined decline of multiple cysteine protease DUBs rather than dysfunction of a single enzyme.

Supplementary Figure 4: Impact of USP7 inhibition on ubiquitylome of iNeurons

(A) Schematic illustrating the K- ϵ -GG antibody-mediated ubiquitylated peptide enrichment and total proteome analysis of iPSC-derived iNeurons treated either with DMSO or 2.5 μ M P5091 (USP7 inhibitor) for 24 hours (N = 3). (B) Representative images showing the morphology of 14 days post-differentiated iNeurons after treatment with DMSO or 2.5 μ M P5091 (USP7 inhibitor) for 24 hours (magnification = 20X; scale = 200 μ m; repeated across N = 3). (C) PCA of proteome changes in iNeurons (N = 3). (D) PCA of ubiquitylome changes in iNeurons (N = 3). Ellipses represent 95% confidence intervals for highlighted drug treatments in both plots. Percent variance explained by each principal component is indicated (N = 3). (E) Volcano plot of protein abundance changes in USP7-inhibited neurons (N = 3). (F) Volcano plot of ubiquitylated peptide changes in USP7-inhibited neurons (N = 3). In both volcano plots, vivid pink dots indicate significantly altered substrates of USP7 from (Ramirez et al. 2021); deep indigo dots mark substrates of USP7 that are not significantly altered; vivid yellow dots denote other significantly altered proteins or peptides; light grey dots indicate non-significant proteins or peptides. Differential abundance was assessed using Spectronaut (Qvalue). Dashed lines indicate thresholds ($|\text{average log}_2 \text{FC}| > 0.3$; Qvalue < 0.05). (G) Left: Scatter plot comparing differentially enriched ubiquitylated sites between mouse aging (old vs. young; x-axis; from (Marino et al. 2025)) and USP7-inhibited iNeurons (P5091 vs. DMSO; y-axis; this study; N = 3). Right: Quadrant-based Over Representation Analysis (ORA) of the top 10 biological processes. Data includes ubiquitylated site changes with adj.pvals < 0.05 (for mouse) and Qvalue < 0.05 (for iNeurons). (H) Heatmap representing the comparison of average Log₂ FC of all enriched ubiquitylated sites from Quadrant 1 (Q1) in (G) between mouse aging and USP7-inhibited iNeurons. (I) Heatmap representing the comparison of average Log₂ FC of all enriched ubiquitylated sites from Quadrant 3 (Q3) in (G) between mouse aging and USP7-inhibited iNeurons. For both heatmaps, only differentially enriched sites with adj.pvals < 0.05 (for mouse) and Qvalue < 0.05 (for iNeurons) were used. Related to Supplementary Data 3.

7. Figure 5: As an antioxidant, NAC can have multiple intracellular effects and influence various signaling pathways and regulators beyond cysteine protease DUB activity. Therefore, NAC could affect ubiquitination levels and proteasome activity through several mechanisms, making it challenging to establish a direct link between DUB activity, proteasome activity, and aging. The authors should address this in more detail in the discussion.

We thank the reviewer for highlighting this important point. We agree that NACET, as a broad-spectrum antioxidant, can influence multiple intracellular pathways beyond cysteine protease DUBs, which complicates the direct attribution of effects on ubiquitination and proteasome activity solely to DUB restoration. We had already discussed this limitation in the manuscript, and we have now expanded the text to clarify these multiple mechanisms for the reader. The revised discussion is now included in lines 713-716:

"As a broad-spectrum antioxidant and direct ROS scavenger, NACET may also preserve the reduced state of catalytic cysteines in E3 ligases and other redox-sensitive enzymes, thereby indirectly reducing protein misfolding, ubiquitylation burden, and proteasomal stress.

While these findings highlight the potential of redox-based interventions in late life to restore aspects of proteostasis and demonstrate the reversibility of oxidative inactivation, several important limitations remain. NACET treatment in this study was limited to 12 days, and the long-term safety, efficacy, and durability of redox-based interventions in aging brains remain to be determined. Notably, prolonged NACET administration at comparable doses was well tolerated in young animals (Realini et al. 2025), supporting its feasibility; however, whether similar outcomes extend to aged brains requires further investigation. Moreover, although we establish a link between redox-driven DUB inactivation and proteostasis decline, the relationship between altered DUB function and other canonical hallmarks of brain aging, as well as downstream physiological and behavioral consequences, remains to be systematically explored."

Minor points:

1. In line 44, please add "(proteostasis)" after "protein homeostasis."

We thank the reviewer for the suggestion. We have included "proteostasis" and the updated line number is 48, highlighted in blue.

2. Lines 47-49: The group of Ulrich Hartl showed that proteasome activity increases with age in *C. elegans* (Walther et al., Cell 2015, Figure S2c). Thus, a decline in proteasome activity during aging might not happen in all animals. Nevertheless, I agree with the authors that discussing this in detail in the introduction may be not necessary. Instead, the authors could consider rephrasing the sentence as follows: "Alterations in proteasome activity and the ubiquitylome have been well characterized in the context of aging in both invertebrates and vertebrates."

We thank reviewers for raising this valuable point. We have included the suggestions in the introduction, along with correct citations, for readers in lines 51-53.

Reviewer #2 (Remarks to the Author):

Reviewer #3 (Remarks to the Author):

The study by Sahu and colleagues finds that Deubiquitinating enzyme function declines with age in the brains of mice and killifish, that this is driven not by changes in DUB abundance but oxidative state and that ex vivo treatment with antioxidants can rescue these changes. The findings are interesting, the work is technically sophisticated and the claims are impactful. All of which makes the work of interest with an overall moderate impact. However, there are a number of findings in where the claims made by the researchers are not fully supported by their data and additional experiments would be needed to validate these claims.

1. The number of animals in Fig 1 / Fig S1 is very low $N = 3$ and only males. If possible, it would be important for researchers to show that key findings are replicable in a larger cohort and not just the product of chance variation from a small cohort size.

We thank the reviewer for this important suggestion. To address the concern, we replicated the DUB activity experiment in an independent cohort (cohort 2) comprising 3 male mice. The global reduction in DUB activity in this cohort is now shown as a box plot in Fig. S1D.

Furthermore, we have included a comparison of DUB activity from both cohorts alongside DUB abundance in the heatmap presented in Fig. 1F, which allows readers to directly assess the reproducibility of the observed changes and their independence from protein abundance.

We hope these additions clarify and strengthen the original statement. For the reviewer's convenience, we have attached the text from the results section (lines 111-113) and the plot below.

“This decline was independently validated in a second experimental cohort, which revealed reduced activity in 20 out of 54 detected DUBs, thereby confirming the robustness of our findings (Fig. S1D).”

Figure S1D: Boxplot showing DUB enrichment in young (3 months) and old (33 months) mouse brains (cohort 2). 47 DUBs detected in both groups are shown (N = 3; |average log₂ FC| > 0.58; Qvalue < 0.05; Wilcoxon rank-sum test). Data are shown as median with interquartile range.

2. The Researchers claim that age-related changes in DUB activity cannot be fully explained by changes in protein abundance but do not provide a satisfactory demonstration of this (only showing a volcano plot where some DUBs are significantly reduced while others significantly increase with age). If anything, the data that there is significant DUB remodeling with age and would support changes in abundance of DUBs as a driver of these changes.

We thank the reviewer for this valuable comment. To clarify the relationship between DUB activity and protein abundance during aging, we have now included comparative heatmaps (Fig. 1F and Fig. S1F, cited in lines 132-140) that show DUB activity alongside total protein abundance in mouse and killifish brains. DUBs not detected in the total proteomics datasets are indicated as ‘NA’.

These heatmaps clearly demonstrate that age-related changes in DUB activity occur largely independently of changes in protein abundance during brain aging

in both mice and killifish. For a subset of DUBs whose total protein abundance was not detected (7 in mouse and 1 in killifish), it was not possible to assess whether the observed activity changes were independent of protein abundance.

For the reviewer's convenience, we have attached the updated text and heatmaps. Please note that the figure order and layout have been slightly adjusted to improve the logical flow and clarity of presentation.

“Importantly, in both species, the age-dependent decline in DUB activity was largely independent of corresponding changes in DUB protein abundance (Fig. 1F, Fig. S1F). In mouse brains, 20 out of 27 DUBs that exhibited age-dependent changes in activity in at least one cohort showed no significant alteration in protein abundance (absolute \log_2 FC > 0.58 and Pvalue < 0.05) (Fig. 1F). Similarly, in aging killifish brains, 5 out of 6 DUBs displaying reduced activity did not show changes in protein abundance, with the exception of USP25, which exhibited decreased protein levels (Fig. S1F). For a subset of DUBs whose total protein abundance was not detected (7 in mouse and 1 in killifish), it was not possible to assess whether the observed activity changes were independent of protein abundance.”

DUBs enrichment vs. abundance during aging in mice

	Activity (Cohort 1)	Activity (Cohort 2)	Abundance (Mouse TMT)		Activity (Cohort 1)	Activity (Cohort 2)	Abundance (Mouse TMT)
ATXN3	0.02	0.193	0.062	USP16	0.176	NA	NA
BAP1	0.109	0.008	NA	USP19	0.052	0.032	0.012
CYLD	0.158	0.115	0.007	USP2	0.053	0.042	NA
JOSD2	0.127	0.438	0.003	USP20	0.932	0.084	0.001
MINDY1	0.692	0.134	0.002	USP21	0.049	0.036	NA
MINDY2	0.799	0.029	0.746	USP22	0.348	0.088	0.328
MINDY3	0.822	0.038	0.007	USP24	0.604	0.05	0.944
MINDY4	0.017	0.131	NA	USP25	0.041	0.185	0.019
OTUB1	0.093	0.535	0.863	USP28	0.018	0.228	NA
OTUB2	0.013	0.038	0.004	USP3	0.24	0.193	NA
OTUD1	0.579	0.024	NA	USP30	0.399	NA	NA
OTUD3	0.089	NA	NA	USP33	0.78	0.112	0.073
OTUD4	0.427	0.011	0.335	USP34	0.64	0.012	0.943
OTUD5	0.023	0.267	NA	USP38	0.068	0.036	NA
OTUD6B	0.771	0.123	0	USP39	0.226	NA	0.261
OTUD7A	0.009	0.212	0.878	USP4	0.215	0.066	0.044
OTUD7B	0.132	0.024	0.009	USP40	0.172	0.06	NA
OTULIN	NA	NA	0.111	USP42	0.535	0.123	NA
TNFAIP3	NA	0.373	NA	USP45	0.147	0.225	NA
UCL1	0.027	0.03	0.046	USP46	0.351	0.503	0.769
UCL3	0.85	0.912	0.998	USP47	0.007	0.039	0.981
UCL4	NA	0.241	NA	USP48	0.058	0.156	NA
UCL5	0.018	0.037	0.002	USP5	0.142	0.515	0.054
USP10	0.035	0.012	0.585	USP54	0.649	0.808	0.064
USP11	0.099	0.444	0.008	USP7	0.009	0.04	0.019
USP12	0.656	0.065	NA	USP8	0.614	0.966	0.042
USP13	0.411	0.126	0.797	USP9X	0.727	0.013	0.131
USP14	0.041	0.076	0.002	VCPIP1	0.469	0.651	0.171
USP15	0.001	0.001	0.029	YOD1	0.018	0.044	0.482
				ZRANB1	0.104	0.069	NA

Figure 1F: Heatmap comparing age-associated changes in DUB activity (this study; cohort 1: 30 vs. 3 months; cohort 2: 33 vs. 3 months; N = 3) and protein abundance (TMT proteomics from (Marino et al. 2025); 33 vs. 3 months). Values represent Pvalues from unpaired t-tests with Welch's correction. 'NA' indicates DUBs not detected. DUBs highlighted in red exhibit significant (Pvalue < 0.05) age-associated activity changes in at least one cohort.

Figure S1F: Heatmap comparing age-associated changes in DUB activity (this study; 39 vs. 5 wph; N = 3) and protein abundance (DIA proteomics from (Di Fraia et al. 2025)) in killifish brains. Values represent Pvalues from unpaired t-tests with Welch's correction. 'NA' indicates DUBs not detected. DUBs highlighted in red exhibit significant (Pvalue < 0.05) age-associated activity changes.

- Fig S1A it would be important to label names of the DUBs marked in red so readers can compare their change in abundance with the change in activity of these DUBs with age shown in Fig S1C. The researchers should also normalize the prevalence of active DUBS in Fig S1C to the relative abundance of these DUBS in Fig S1A to show that the changes they report are indeed not driven by changes in abundance.

We thank the reviewer for this suggestion. To address this concern, we have now included comparative heatmaps (Fig. 1F and Fig. S1F), as stated in our previous answer, which display DUB activity alongside total protein abundance for all detected DUBs. We believe that these heatmaps allow readers to directly assess the relationship between activity and abundance, clarifying that the age-related changes in DUB activity are largely independent of protein levels.

4. The researchers claim the following factors are enriched but these changes are all non-significant PSMD7, PSMD14, and COPS5 (Fig 1d P=0.15 to P=0.66)

We thank the reviewer for this helpful observation referring to lines 117-120, where it was stated earlier as: “Of note, we also observed the enrichment of metalloprotease DUBs, such as PSMD7, PSMD14, and COPS5, among others, due to their physical interaction with cysteine-dependent DUBs. However, their enrichment showed no significant difference between young and old (Fig. S1E).” We initially included the statement to highlight the specificity and robustness of our enrichment toward cysteine-dependent DUBs, rather than metalloprotease DUBs.

We acknowledge that PSMD7, PSMD14, and COPS5 did not show statistically significant enrichment. These proteins belong to the JAMM family of metalloprotease deubiquitylases, which lack a catalytic cysteine residue. Because the activity-based probes used in this study selectively target cysteine-dependent DUBs, JAMM-type enzymes are not directly labeled by the probe.

Their apparent enrichment most likely reflects co-purification through physical interactions with probe-labeled cysteine DUBs, rather than true probe reactivity. For instance, PSMD7, a metalloprotease component of the 26S proteasome lid, interacts with the cysteine DUB USP14 (Ramirez et al. 2021).

To prevent possible misinterpretation, we have removed this sentence from the main text. The corresponding data remain available in Supplementary Table 2 for readers interested in a complete dataset.

5. The researcher claim that declines in DUB activity precede impaired proteasome function but this seems to be based just on chymotrypsin-like activity. This measure is going to record not just 26S activity (which is relevant to the researchers question) but also 20S (whose levels some researchers find increases with age in response to diminished 26S) and also immunoproteasome which may increase with age driven by inflammatory signalling. As such this claim is potentially misleading the researchers should do native PAGE activity blots or similar to look at the activity of 26S in their samples.

We thank the reviewer for raising this important point regarding the interpretation of proteasome activity measurements. We agree that the chymotrypsin-like (CT-L) assay does not distinguish between the activities of the 20S, 26S, and immunoproteasome complexes, which could potentially lead to an overgeneralized interpretation. To address this, we have now explicitly clarified this limitation in the Methods section (lines 876-878) with the following statement: “This assay reflects the combined activity of multiple proteasome species,

including 20S, 26S, and immunoproteasome complexes, rather than selectively measuring 26S activity.”

In addition, for better clarity to readers, we have incorporated the following text in the Discussion (lines 696–701): “Although the CT-L assay used here reflects the combined activity of multiple proteasome species, coordinated declines in both 20S and 26S proteasome activity have been reported in aging brains (Dasuri et al. 2009; Burov et al. 2023; Parker et al. 2025). Nonetheless, we acknowledge that age-related compensatory mechanisms, such as an increased 20S/26S activity ratio or elevated immunoproteasome function, may mask or offset a more pronounced decline in 26S proteasome activity (Kelmer Sacramento et al. 2020; Rao et al. 2024).”

To further support this interpretation, we have summarized below key literature reporting declines in 20S, 26S, and immunoproteasome activities during brain aging, for the reviewer’s convenient reference:

- (1) (Parker et al. 2025) reported a significant reduction in both 20S ($\approx 70\%$) and 26S ($\approx 50\%$) proteasome activities in mouse brain (24 months vs. 12 months) using a native-PAGE activity assay with the MV151 fluorescent probe to label active proteasome centers. Interestingly, while they observed an increase in 26S proteasome assembly and a modest decline in 20S assembly, the overall chymotrypsin-like activity per complex was markedly reduced with age (Figure A-D). Similar findings were reported in the hippocampus, a region critical for learning and memory.
- (2) (Dasuri et al. 2009) also observed a significant decrease in both 20S and 26S proteasome activities in aged rat brain (25 months vs. 3 months) using proteasome complexes purified via Fast Protein Liquid Chromatography (FPLC) (Figure E).
- (3) Finally, (Rao et al. 2024) demonstrated a significant reduction in 20S proteasome activity in 24-month-old mouse cortex, which they attributed primarily to impaired turnover of 20S core subunits rather than reduced proteasome abundance (see Fig. 4 and EV4 in their study).

Together, these studies demonstrate that both 20S and 26S proteasome activities decline during brain aging. Consequently, the reduction in CT-L activity in our samples robustly reflects a global decrease in proteasome catalytic capacity.

Given the available literature and the fact that a new set of aged experimental animals would be required to perform the suggested native PAGE activity assay, we have opted not to conduct these additional experiments. Instead, we will

provide a more precise statement of the interpretation and limitations of our current data, as indicated above.

[Figures Redacted]

Figure legends from referred publications: (A) Native-PAGE blot of whole-brain samples from young (12 ± 1 Mo, N = 10) and old (24 ± 2 Mo, N = 11) mice showing active proteasome cores (top), proteasome assembly via anti-PSMB5 native-PAGE blot (middle), and total protein by silverstain (bottom). (B–D) Quantification of proteasome activity and assembly. (E) Rat brain lysates were fractionated on Superose- 6 HR 10/300 gel filtration chromatography column and 500 μ l fractions were collected as described in methods section. Fractions were then assayed for proteasome activity using Succinyl-Leu-leu-Val-Tyr-7-amido-4-methylcoumarin as substrate under conditions optimized for 26S proteasomes (+ATP) and for 20S proteasomes (+0.02% SDS). (F-G) The distribution of chymotrypsin-like activity and activity of the β 5i subunit between the 20S and 26S proteasomes in cerebral cortex lysates of C57BL/6 mice of different ages (n = 4 for each group). (F) The proteasome activity in the gel regions corresponding to the mobility of 20S and 26S proteasomes in native PAGE was determined using the corresponding fluorogenic substrates. (G) The chymotrypsin-like activity and the activity of the β 5i subunit were assessed via analysis of the optical density of the signals in the (F) using the ImageJ software. Results are presented as the mean \pm SD.

Reviewer #4 (Remarks to the Author):

This manuscript investigates the regulatory mechanisms of deubiquitinating enzymes (DUBs) activity in the context of brain aging in vertebrates. It establishes that DUBs activity significantly declines with age in two vertebrate models, independent of protein abundance. The study elucidates that oxidative stress-mediated thiol oxidation is a pivotal mechanism underlying the decline in DUBs activity, a process that is reversible. Furthermore, it reveals that the reduction in DUB activity occurs prior to the impairment of proteasome function, thereby providing new evidence for the early molecular events associated with protein homeostasis imbalance during brain aging. The manuscript also validates the rescue effect of redox regulation on molecular disorders related to brain aging through interventions using NACET.

However, the current version of the manuscript contains several shortcomings that require attention.

Major Points :

1. The manuscript's mixed use of experimental animal strains and genders may introduce systematic errors: DUB activity assays were performed using male C57BL/6J mice (line-130), male C57BL/6N mice were used in Figure 4, and female C57BL/6J mice were used for NACET treatment (Figure 5). However, no evidence was provided to indicate: 1) the basis for the consistency in brain aging-related molecular characteristics (such as ROS levels and DUB activity) between C57BL/6J and C57BL/6N; 2) whether the pattern of DUB activity changes exhibits sex differences, and whether the rescuing effect of NACET is applicable only to females.

We thank the reviewer for raising this important point regarding potential strain- and sex-specific differences in brain aging phenotypes. To directly address this concern, we conducted an additional series of experiments using female mice from the C57BL/6J and C57BL/6N substrains. We quantified DUB activity and total thiol levels, along with proteasome activity and total ubiquitylated proteins across these groups, and observed highly similar age-associated changes regardless of sex or substrain. These new results are now incorporated into Fig. 1A, Fig. 2A, and Fig. 4B-E, demonstrating that the key molecular features described in this study, redox imbalance and loss of DUB activity, are consistent across both sexes and both C57BL/6 substrains.

Regarding the NACET treatment experiment, we agree that sex-specific effects cannot be entirely excluded. However, extending the NACET intervention to male mice is beyond the scope of the present study, which aimed primarily to establish proof of concept for the reversibility of age-associated thiol oxidation and DUB inactivation *in vivo*. Notably, (Realini et al. 2025) (Figure 5 of the original manuscript) reported that NACET treatment in young (2–4 months) and old (18–25 months) C57BL/6J male mice partially rescued the transcriptional aging phenotype of the retina, suggesting that NACET effectively crosses the blood-retinal barrier and reverses aging signatures in males as well. This provides supportive evidence that the compound is bioavailable to central nervous system tissues in both males and females.

Importantly, because the age-related decline in thiol levels and DUB activity was comparable in males and females, the mechanisms targeted by NACET appear not to be sex-specific. Future studies will be required to fully explore potential sex-dependent differences and long-term physiological consequences in therapeutic response.

Figure 1A: Top: schematic illustrating the principle of fluorescent substrate–based measurement of deubiquitylating enzyme (DUB) activity. Bottom: DUB activity in brain lysates from young (3 months) and old (30 months) C57BL/6J male mice (left, N = 4) and young (3 months) and old (22–24 months) C57BL/6J female mice (right, N = 5). Unpaired t-test with Welch’s correction. RFU = relative fluorescence units. Data are shown as mean \pm SD.

Figure 2A: Top: schematic illustrating Ellman's reagent (DTNB)-based detection of reduced thiol groups, producing the yellow TNB product measurable at 412 nm via spectrophotometry. Bottom: reduced thiol concentrations in brain lysates from young (3 months) and old (30 months) C57BL/6J male mice (left, N = 8) and young (3 months) and old (22–24 months) C57BL/6J female mice (right, N = 5). Unpaired t-test with Welch's correction. RFU = relative fluorescence units. Data are shown as mean ± SD.

Figure 4: (A) Schematic illustrating the temporal study of molecular changes during aging in C57BL/6N male and female mouse brains (N = 3-5). (B) Reduced thiol concentrations in mouse brains across age groups (N = 5 for all male and female groups, except N = 4 for 30 months old females). (C) DUB activity across age groups (N = 3 for 30 months old females; N = 4 for 3 months old males and 6 and 18 months old females; N = 5 for other groups). (D) Total ubiquitylated protein levels (N = 3 for females; N = 4 for males). (E) Proteasome activity (N = 3 for 24 months old males; N = 4 for other male groups and 30 months old females; N = 5 for other female groups).

2. The results of DUB activity confirmed that "the activity of UCHL1 and YOD1 decreased, while the activity of CYLD and USP9X remained unchanged" (Fig S1D), but only four examples of DUBs were presented, and no enrichment analysis results of all differentially active DUBs were provided. This weakens the specificity of the conclusion that cysteine protease DUBs are sensitive to oxidative stress.

We thank the reviewer for this valuable comment. To clarify the relationship between DUB activity and protein abundance during aging, we have now included comparative heatmaps (Fig. 1F and Fig. S1F, cited in lines 132-140) that show DUB activity alongside total protein abundance in mouse and killifish brains. DUBs not detected in the total proteomics datasets are indicated as 'NA'.

These heatmaps clearly demonstrate that age-related changes in DUB activity occur largely independently of changes in protein abundance during brain aging in both mice and killifish.

For the reviewer's convenience, we have attached the updated text and heatmaps. Please note that the figure order and layout have been slightly adjusted to improve the logical flow and clarity of presentation.

"Importantly, in both species, the age-dependent decline in DUB activity was largely independent of corresponding changes in DUB protein abundance (Fig. 1F, Fig. S1F). In mouse brains, 20 out of 27 DUBs that exhibited age-dependent changes in activity in at least one cohort showed no significant alteration in protein abundance (absolute \log_2 FC > 0.58 and Pvalue < 0.05) (Fig. 1F). Similarly, in aging killifish brains, 5 out of 6 DUBs displaying reduced activity did not show changes in protein abundance, with the exception of USP25, which exhibited decreased protein levels (Fig. S1F). For a subset of DUBs whose total protein abundance was not detected (7 in mouse and 1 in killifish), it was not possible to assess whether the observed activity changes were independent of protein abundance."

DUBs enrichment vs. abundance during aging in mice

Figure 1F: Heatmap comparing age-associated changes in DUB activity (this study; cohort 1: 30 vs. 3 months; cohort 2: 33 vs. 3 months; N = 3) and protein abundance (TMT proteomics from (Marino et al. 2025); 33 vs. 3 months). Values represent Pvalues from unpaired t-tests with Welch's correction. 'NA' indicates DUBs not detected. DUBs highlighted in red exhibit significant (Pvalue < 0.05) age-associated activity changes in at least one cohort.

DUBs enrichment vs. abundance during aging in killifish

Figure S1F: Heatmap comparing age-associated changes in DUB activity (this study; 39 vs. 5 wph; N = 3) and protein abundance (DIA proteomics from (Di Fraia et al. 2025)) in killifish brains. Values represent Pvalues from unpaired t-tests with Welch’s correction. ‘NA’ indicates DUBs not detected. DUBs highlighted in red exhibit significant (Pvalue < 0.05) age-associated activity changes.

- Oxidative stress was indirectly suggested through "changes in antioxidant enzyme expression" (Supplementary Figure S2A) and "decreased thiol concentration" (Figure 2A), but without directly detecting markers of direct oxidative damage such as ROS levels, MDA, or protein carbonylation levels in the aged brain, the possibility that thiol oxidation is caused by metabolic disorders rather than ROS cannot be ruled out.

We appreciate the reviewer’s insightful comment. We agree that direct measurements of oxidative damage markers such as ROS, MDA, or protein carbonyls would further strengthen the evidence for oxidative stress. In our study, the interpretation was based on consistent alterations in antioxidant enzyme expression (Supplementary Fig. S2A) and decreased total thiol concentration (Fig. 2A), both of which are well-established indirect indicators of redox imbalance and oxidative stress in the aging brain (Sohal and Orr 2012; Maher 2005; Jones 2006). In addition, in a new set of experiments, we also observed a decline in nuclear factor erythroid 2-related factor 2 (NRF2), the master regulator of antioxidant gene expression and glutathione (GSH) biosynthesis, accompanied by reduced GSH levels in aging mouse brains (Fig. S2B-C). A

similar decrease in NRF2 and GSH levels was observed in rat livers during aging (Suh et al. 2004).

Although direct detection of oxidative damage products was beyond the scope of this study, (Maher 2005) compiled multiple studies reporting alterations in glutathione levels during aging across different model organisms. Consistent with these findings, increased levels of cystine (oxidized cysteine), cysteine-glutathione disulfide, and 4-hydroperoxy-2-nonenal (4-HPNE), along with decreased total glutathione levels, were observed in a mouse brain metabolome dataset (Ding et al. 2021). Moreover, several independent studies have reported elevated ROS production in aged brains (Ali et al. 2006; Cardozo-Pelaez et al. 1999; Forster et al. 1996). Taken together, these data indicate an imbalance in redox homeostasis consistent with increased oxidative stress in the aged brain; however, we acknowledge that a secondary contribution from metabolic dysregulation cannot be entirely ruled out.

To address this comment and clarify our interpretation, we have incorporated this discussion in the revised manuscript in lines 629-639 as follows: “We confirmed the presence of multiple biomarkers indicative of redox imbalance in aged mouse brains, including reduced total thiol content, decreased NRF2 protein levels and GSH concentrations, as well as compensatory alterations in antioxidant enzyme expression, collectively consistent with elevated oxidative stress (Sohal and Orr 2012; Maher 2005; Jones 2006; Suh et al. 2004). These data complement numerous previous studies that reported elevated ROS production in aged brains (Ali et al. 2006; Cardozo-Pelaez et al. 1999; Forster et al. 1996), and an age-dependent increase of metabolites such as cystine (oxidized cysteine), cysteine-glutathione disulfide, and 4-hydroperoxy-2-nonenal (4-HPNE), indicative of glutathione metabolism dysregulation during aging (Ding et al. 2021; Maher 2005). While redox imbalance likely represents a key contributing factor driving the loss of DUB function in the aging brain, a potential role of broader metabolic alterations and other post-translational modes of DUB modulation, such as phosphorylation, ubiquitylation, or protein-protein interactions, cannot be fully excluded (Snyder and Silva 2021).”

4. NACET treatment only validated the rescue at the molecular level, and it was unable to prove that the molecular-level improvements could translate into the restoration of brain function; furthermore, the GSH levels in the brain after NACET treatment were not detected, making it impossible to confirm whether "thiol restoration" was achieved through GSH-mediated redox balance reconstruction.

We thank the reviewer for bringing this important point to our attention. To directly address the concern regarding thiol restoration and its relationship to glutathione-dependent redox balance, we quantified glutathione (GSH) levels together with NRF2 protein abundance in young, old, and NACET-treated old mouse brains. This analysis was motivated by the fact that, once inside the cell, NACET is readily de-esterified and deacetylated to supply cysteine. Cysteine can act directly as an antioxidant and can also promote activation of the NRF2 pathway. Moreover, as the rate-limiting precursor for GSH biosynthesis, cysteine supports the production of this major cellular antioxidant.

Consistent with aging-associated redox dysregulation, we observed a reduction in both NRF2 (Fig. S2B) and GSH (Fig. S2C) levels in aged brains compared to young controls. In line with previous reports (Realini et al. 2025), NRF2 protein abundance was restored in NACET-treated old animals. In contrast, GSH levels remained unchanged following NACET treatment, indicating that the observed thiol restoration and recovery of DUB activity occur independently of global GSH-mediated redox balance.

Accordingly, we have revised the Results and Discussion sections to reflect these findings (lines 209–212, 540–549, and 624–625). For the reviewer's convenience, we have included snapshots of the corresponding results below.

While evaluating whether NACET-mediated molecular improvements translate into functional restoration of brain physiology in aged animals would be highly informative, we consider such analyses fall beyond the scope of the current study and represent an important direction for future work.

Figure S2: (B) NRF2 protein levels (see Fig. S6A for the blot) in brain lysates from young (3 months, N = 3) and old (22–24 months, N = 3) C57BL/6J female mice. Unpaired t-test with Welch's correction. (C) GSH concentrations in brain lysates from young (3 months, N = 4) and old (22–24

months, N = 3) C57BL/6J female mice. Unpaired t-test with Welch's correction. Data are shown as mean \pm SD.

Figure S6: (A) Upper: Immunoblot of NRF2 protein levels in young (3 months), old (22-24 months), and old C57BL/6J female mice treated with NACET. Lower: NRF2 protein abundance quantification (lower band) in NACET vs. vehicle-treated aged mouse brains. Total protein levels were assessed using Tubulin (N = 3; unpaired t-test with Welch's correction). (B) Glutathione (GSH) concentrations in NACET vs. vehicle-treated aged mouse brains (N = 3; unpaired t-test with Welch's correction). Data are shown as mean \pm SD.

- The discussion did not address the key limitations: 1) NACET treatment lasted only 12 days, and the safety and efficacy of long-term intervention remain unknown; 2) the association between changes in DUBs activity and other aging markers was not explored.

We thank the reviewer for highlighting these important limitations. We have now explicitly addressed them in the Discussion (lines 713-716) as follows:

“As a broad-spectrum antioxidant and direct ROS scavenger, NACET may also preserve the reduced state of catalytic cysteines in E3 ligases and other redox-sensitive enzymes, thereby indirectly reducing protein misfolding, ubiquitylation burden, and proteasomal stress.

While these findings highlight the potential of redox-based interventions in late life to restore aspects of proteostasis and demonstrate the reversibility of oxidative inactivation, several important limitations remain. NACET treatment in this study was limited to 12 days, and the long-term safety, efficacy, and durability of redox-based interventions in aging brains remain to be determined. Notably, prolonged NACET administration at comparable doses was well tolerated in young animals (Realini et al. 2025), supporting its feasibility; however, whether similar outcomes extend to aged brains requires further investigation. Moreover,

although we establish a link between redox-driven DUB inactivation and proteostasis decline, the relationship between altered DUB function and other canonical hallmarks of brain aging, as well as downstream physiological and behavioral consequences, remains to be systematically explored.”

Minor Points :

1. The molecular weights of the bands in the Western blots results of image S4D were not labeled.

We thank the reviewer for bringing the missing marker to our attention. We have now updated the Western blot image of Figure S4D. For your quick reference, find the attached image below:

Figure S5D: Immunoblot validation of total ubiquitylated protein levels in iNeurons following treatment with different concentrations of PR619 and 10 nM Bortezomib. Total protein levels were assessed using Ponceau staining.

2. PR619, NACET, DUB probes, etc., require the addition of supplier and product numbers to facilitate the replication of related experiments by subsequent researchers.

We thank the reviewer for pointing out the missing information about NACET in the Methods section. We included the information under the heading “NACET treatment in aged mice” in line 785-786, and it is highlighted in yellow for quick reference.

NACET - C7H13NO3S, Advanced ChemBlocs, O32426, Lot #AC99952A; Line 785-786

Information about all other chemicals used is provided in the Methods section.

PR619: Sigma (662141); Line 983

DUB Probes: Biotin-Ahx-Ub-VME (UbiQ-054), Biotin-Ahx-Ub-PA (UbiQ-076),
and Biotin-Ahx-Ub-VS (UbiQ-188); Line 813-814

References

- Ali, Sameh S., Chengjie Xiong, Jacinta Lucero, M. Margarita Behrens, Laura L. Dugan, and Kevin L. Quick. 2006. "Gender Differences in Free Radical Homeostasis during Aging: Shorter-Lived Female C57BL/6 Mice Have Increased Oxidative Stress." *Aging Cell* 5 (6): 565–574.
- Burov, A. V., S. Yu Funikov, T. M. Astakhova, et al. 2023. "Dynamic Changes in the Activities and Contents of Particular Proteasome Forms in the Cerebral Cortex of C57BL/6 Mice during Aging." *Molecular Biology* 57 (5): 897–904.
- Cardozo-Pelaez, F., S. Song, A. Parthasarathy, C. Hazzi, K. Naidu, and J. Sanchez-Ramos. 1999. "Oxidative DNA Damage in the Aging Mouse Brain." *Movement Disorders: Official Journal of the Movement Disorder Society* 14 (6): 972–980.
- Chauhan, Dharminder, Ze Tian, Benjamin Nicholson, et al. 2012. "A Small Molecule Inhibitor of Ubiquitin-Specific Protease-7 Induces Apoptosis in Multiple Myeloma Cells and Overcomes Bortezomib Resistance." *Cancer Cell* 22 (3): 345–358.
- Dasuri, Kalavathi, Le Zhang, Philip Ebenezer, Ying Liu, Sun Ok Fernandez-Kim, and Jeffrey N. Keller. 2009. "Aging and Dietary Restriction Alter Proteasome Biogenesis and Composition in the Brain and Liver." *Mechanisms of Ageing and Development* 130 (11-12): 777–783.
- Di Fraia, Domenico, Antonio Marino, Jae Ho Lee, et al. 2025. "Altered Translation Elongation Contributes to Key Hallmarks of Aging in the Killifish Brain." *Science (New York, N.Y.)* 389 (6759): eadk3079.
- Ding, Jun, Jian Ji, Zachary Rabow, et al. 2021. "A Metabolome Atlas of the Aging Mouse Brain." *Nature Communications* 12 (1): 6021.
- Forster, M. J., A. Dubey, K. M. Dawson, W. A. Stutts, H. Lal, and R. S. Sohal. 1996. "Age-Related Losses of Cognitive Function and Motor Skills in Mice Are Associated with Oxidative Protein Damage in the Brain." *Proceedings of the National Academy of Sciences of the United States of America* 93 (10): 4765–4769.
- Jones, Dean P. 2006. "Redefining Oxidative Stress." *Antioxidants & Redox Signaling* 8 (9-10): 1865–1879.
- Kelmer Sacramento, Erika, Joanna M. Kirkpatrick, Mariateresa Mazzetto, et al. 2020. "Reduced Proteasome Activity in the Aging Brain Results in Ribosome Stoichiometry Loss and Aggregation." *Molecular Systems Biology* 16 (6): e9596.
- Maher, Pamela. 2005. "The Effects of Stress and Aging on Glutathione Metabolism." *Ageing Research Reviews* 4 (2): 288–314.
- Marino, Antonio, Domenico Di Fraia, Diana Panfilova, et al. 2025. "Aging and Diet Alter the Protein Ubiquitylation Landscape in the Mouse Brain." *Nature Communications*

16 (1): 5266.

- Parker, Danitra, Kanisa Davidson, Pawel A. Osmulski, Maria Gaczynska, and Andrew M. Pickering. 2025. "Proteasome Augmentation Mitigates Age-Related Cognitive Decline in Mice." *Aging Cell* 24 (3): e14492.
- Ramirez, Juanma, Gorka Prieto, Anne Olazabal-Herrero, et al. 2021. "A Proteomic Approach for Systematic Mapping of Substrates of Human Deubiquitinating Enzymes." *International Journal of Molecular Sciences* 22 (9): 4851.
- Rao, Nalini R., Arun Upadhyay, and Jeffrey N. Savas. 2024. "Derailed Protein Turnover in the Aging Mammalian Brain." *Molecular Systems Biology* 20 (2): 120–139.
- Realini, Giulia, Rosario Amato, Mahdi Rasa, et al. 2025. "N-Acetyl-L-Cysteine Ethyl Ester (NACET) Induces the Transcription Factor NRF2 and Prevents Retinal Aging and Diabetic Retinopathy." *Redox Biology* 88 (103914): 103914.
- Snyder, Nathan A., and Gustavo M. Silva. 2021. "Deubiquitinating Enzymes (DUBs): Regulation, Homeostasis, and Oxidative Stress Response." *The Journal of Biological Chemistry* 297 (3): 101077.
- Sohal, Rajindar S., and William C. Orr. 2012. "The Redox Stress Hypothesis of Aging." *Free Radical Biology & Medicine* 52 (3): 539–555.
- Suh, Jung H., Swapna V. Shenvi, Brian M. Dixon, et al. 2004. "Decline in Transcriptional Activity of Nrf2 Causes Age-Related Loss of Glutathione Synthesis, Which Is Reversible with Lipoic Acid." *Proceedings of the National Academy of Sciences of the United States of America* 101 (10): 3381–3386.
- Weinstock, Joseph, Jian Wu, Ping Cao, et al. 2012. "Selective Dual Inhibitors of the Cancer-Related Deubiquitylating Proteases USP7 and USP47." *ACS Medicinal Chemistry Letters* 3 (10): 789–792.